

# Large-scale Dynamics of Tropical Cyclone Formation Associated with the ITCZ Breakdown

Chanh Kieu[1,*], Quan Wang[2], and The-Anh Vu[1]

[1]Department of Earth and Atmospheric Sciences, Indiana University, Bloomington, IN 47405
[2]Department of Mathematics, Sichuan University, Sichuan, China

**Correspondence:** ckieu@indiana.edu

**Abstract.** This study examines the formation of tropical cyclones (TC) from the large-scale perspective. Using the nonlinear dynamical transition framework recently developed by Ma and Wang, it is shown that the large-scale formation of TCs can be understood as a result of the Principle of Exchange of Stabilities in the barotropic model for the Intertropical Convergence Zone (ITCZ). Analyses of the transition dynamics at the critical point reveal that the maximum number of TC disturbances that the Earth's tropical atmosphere can support at any instant of time has an upper bound, which is $\sim 12$ for the current atmospheric condition. Additional numerical estimation of the transition structure on the central manifold of the ITCZ model confirms this important finding, which offers an explanation for a fundamental question of why the Earth's atmosphere can support a limited number of TCs globally each year.

## 1 introduction

The life cycle of a tropical cyclone (TC) is typically divided into several stages including early genesis, tropical disturbance, tropical depression, tropical storm, hurricane, and finally the dissipation. Among these five stages, the tropical cyclogenesis (TCG), defined as a period during which a weak atmospheric disturbance grows into a mesoscale tropical depression with a close isobar and the maximum surface wind > 17 m s$^{-1}$ (Karyampudi and Pierce, 2002; Tory and Montgomery, 2006), is perhaps the least understood due to its unorganized structure as well as ill-defined characteristics of TCs. During this TCG period (typically 2-5 days), synergetic interactions among various dynamical and thermodynamic processes at different scales can result in an eventually self-sustained, warm-core vortex before the subsequent intensification can take place. These early formation processes are so intricate that no single or distinct mechanism could operate for all TCs, rendering the TCG forecasting very challenging in practice. The multi-facet nature of TCG is the main factor preventing us from obtaining a complete understanding of TC formation and development at present.

Early studies by Gray (1968, 1982) provided a list of necessary climatological conditions for TCG to occur, which include: (i) an underlying warm sea surface temperature (SST) of at least 26$^o$C; (ii) a finite-amplitude low-level cyclonic disturbance;





(iii) weak vertical wind shear; (iv) a tropical upper tropospheric trough; and (v) a moist lower to middle troposphere. While the above conditions for TCG have been well documented in numerous observational and modeling studies since then, an elusive issue is that the actual number of TCs composes a small fraction of the cases that meet all these necessary conditions in the tropical region every year. Moreover, TCG varies wildly among different ocean basin due to relative importance of large-scale

disturbances, local forcings, and surface conditions, thus inheriting strong regional characteristics that common criteria may not be applied everywhere. For example, TCG in the North Atlantic basin often show strong connection to active tropical waves originated from the South African Jet (Avila and Pasch, 1992; DeMaria, 1996; Molinari et al., 1999). In the northwestern Pacific basin, studies by Yanai (1964); Gray (1968, 1982); Mark and J. (1993); Ritchie and Holland (1997); Harr et al. (1996); Nakato et al. (2010) showed that the genesis is mostly related to Intertropical Convergence Zone (ITCZ) and monsoon activities.

In the northeastern Pacific, vortex interaction associated with the topographic and tropical waves seems to generate abundant disturbances that act as the seeds of TCG (Zehnder et al., 1999; Molinari et al., 1997; Wang and Magnusdottir, 2006; Halverson et al., 2007; Kieu and Zhang, 2010).

Other large-scale conditions that can interfere with TCG have been also reported in previous studies such as the Saharan air layer (SAL; Dunion and Velden (2004)), upper-level potential vorticity (PV) anomalies (Molinari and Vollaro, 2000; Davis

and Bosart, 2003), mixed Rossby-gravity waves Aiyyer and Molinari (2003), the ITCZ breakdown (Ferreira and Schubert, 1997; Wang and Magnusdottir, 2006), or multiple vortex merges (Simpson et al., 1997; Ritchie and Holland, 1997; Wang and Magnusdottir, 2006; Kieu and Zhang, 2008; Kieu, 2015). Along with this diverse nature of TCG in different basins, observational and modeling studies of TC development have shown that the evolution of tropical disturbances during the early genesis stage often encompasses a wide range of scales from convective-scale hot towers, mesoscale convective systems, to

large-scale quasi-balanced lifting and cloud-radiation feedbacks (e.g., Riehl and Malkus, 1958; Yanai, 1964; Gray, 1968; Zhang and Bao, 1996; Ritchie and Holland, 1997; Simpson et al., 1997). In this regard, TCG is a truly multi-scale process that relative importance of different mechanisms must be carefully examined when studying the TCG in real atmospheric conditions.

Recent effort in the TCG research has been shifted from examining local mechanisms to a broader perspective of how environmental conditions can produce and maintain TC disturbances during TC early development (Wang and Magnusdottir,

2006; Dunkerton et al., 2009; Montgomery et al., 2010; Wang et al., 2012; Lussier et al., 2013; Zhu et al, 2015; Wu and Shen, 2016). The most current attempt in quantifying the large-scale factors governing the TCG in the North Atlantic basin focuses on the so-called "pouch" conceptual model, which treats an early TC embryo as a protected region within large-scale easterly waves (Wang et al., 2010, 2012; Dunkerton et al., 2009; Montgomery et al., 2010). To some extent, this pouch idea can be considered as an advance of the requirement of an incipient disturbance for TCG to occur that was originally put forth by Gray

(1968). Much of the development along this "pouch" idea has been on tracking wave packets in the co-moving frame required to protect the mid-level disturbances (the so-called Kelvin cat-eye in Dunkerton et al. (2009); Lussier et al. (2013)).

Despite much progress over last decades, several outstanding issues in the TCG study still remain. From the global perspective, a particular question of what is the maximum number of TCs that the Earth's tropical atmosphere can form and support in any given day has not been adequately addressed. Answering this question will help explain a long-standing question of why

the Earth has only a specific number of ∼ 100 TCs globally every year. A recent modeling study of the global TC formation by





Kieu et al. (2018) demonstrated indeed that the daily number of TCG events is intriguingly bounded (<10), even in a perfect environment. This number is quite consistent with a simple scale analysis based on the typical scale of TCs with a diameter $\sim$ 3000-km, which shows that there should have less than 14 TCs on the Earth's atmosphere at any given time, assuming that the radius of the Earth is $\sim$6400 km. Using idealized simulations for a tropical channel, Kieu et al. (2018) showed in fact that TCG occurs in episodes of 7-10 storms each time with a frequency between the episodes of 12-16 days (Figure 1). This episodic development at the global scale as well as the upper bound of $\sim$ 10 storms for each episode as obtained from these idealized experiments suggests that there must have some large-scale environmental conditions or intrinsic properties of the tropical dynamics, which control the TCG processes beyond the basin-specific mechanisms examined in previous studies.

While recent advance in global numerical models can reasonably capture the very early stage of the TCG and serve as guidance for operational TCG forecasts, analytical models of TC development have been confined mostly to the later stage of TC development such that the axisymmetric characteristics of disturbances could be employed. The axisymmetry is critical for the theoretical purposes, because it reduces the Navier-Stokes equations to a set of approximated equations for which some balance constraints and simplifications can be employed.

Given the various basin-specific mechanisms that could produce TCs beyond the axisymmetric model for an individual TC, the main objective of this study is to focus specifically on a large-scale mechanism behind the formation of tropical disturbances associated with the ITCZ breakdown. This special pathway is very typical at the global scales whereby converging winds from the two hemispheres could set up a right environment for large-scale stability to develop (Gray, 1968; Yanai, 1964; Zehnder et al., 1999; Molinari et al., 2000; Ferreira and Schubert, 1997; Wang and Magnusdottir, 2006). Indeed, satellite observations often show that the ITCZ tends to undulate and break into a series of mesoscale vorties or disturbances, some of which may eventually grow into TCs (Agee, 1972; Hack et al., 1989; Ferreira and Schubert, 1997). This is especially apparent in the WPAC basin where early studies by Gray (1968, 1982) showed that TCG primarily occurs along the ITCZ, which accounts for nearly 80 percent of TCG occurrences in this area.

Although the ITCZ breakdown appears to be a slow process as compared to other pathways such as vortex merger (e.g., Wang and Magnusdottir, 2006; Kieu and Zhang, 2008, 2010) or tropical easterly waves (e.g., Zehnder et al., 1999; Molinari et al., 1997; Halverson et al., 2007; Dunkerton et al., 2009; Montgomery et al., 2010; Wang et al., 2012), it is an inherent property of the tropical atmosphere at the global scale that could provide a source of large-scale disturbances responsible for TCG. To minimize the complication due to the basin-specific features, we thus limit our study of the global TC formation to an idealized aqua-planet configuration to facilitate the analytical analyses in this study.

The rest of the paper is organized as follows. In the next section, an analytical model for the large-scale TCG based on the ITCZ breakdown model is presented. Section 3 presents detailed analyses of the principle of exchange of stabilities for the ITCZ model as well as the stability analyses of the dynamical transition. Numerical examination will be discussed in Section 4, and concluding remarks are given in the final section.



## 2 Formulation

A unique characteristic of the ITCZ that provides a favorable environment for TCG is the highly unstable zone along the ITCZ where easterly and westerly winds from the two hemispheres converge. Such a zone with strong horizontal shear is well documented along the tropical belt where the potential vorticity gradient changes sign, providing a necessary condition for

disturbances to grow (Charney and Stern, 1962; Ferreira and Schubert, 1997). Thus, a disturbance embedded within in the ITCZ can trigger a nonlinear growth and extract the energy from the background, resulting in a potential amplification of the disturbance.

Because of such a dominant role of the ITCZ in the global TC formation, a natural model for TCG should take into account the large-scale ITCZ breakdown processes. This ITCZ breakdown model is particularly suitable for an aqua-planet that does

not have other triggering mechanisms such as land-sea interaction or terrain effects. As a result, we will consider the ITCZ breakdown as a starting model for the TCG at the global scale examined in this study. Inspired by the modeling studies of the ITCZ breakdown based on the shallow water equation by Ferreira and Schubert (1997), we examine a similar model for the ITCZ dynamics on a horizontal plane in which the governing equation for the ITCZ can be reduced to an equation for the conservation of potential vorticity as follows

$$\frac{d\Delta\psi}{dt} = \nu_e \Delta^2 \psi + F - \alpha\Delta\psi - \beta\frac{\partial\psi}{\partial x}, \tag{1}$$

where the horizontal streamfunction $\psi$ has been introduced as a result of the continuity equation, $\nu_e$ is horizontal eddy viscosity, $\alpha$ is a relaxation time, and $F$ is an external force that represents the either a source/sink of mass within the ITCZ or vorticity source [1]. As discussed in Ferreira and Schubert (1997), the mass source/sink term $F$ is important for the ITCZ dynamics, because the horizontal dynamics could not fully capture the vertical mass flux within the ITCZ. Unlike the original ITCZ

model in Ferreira and Schubert (1997), we have however introduced in the above model (1) an explicit drag forcing term to represent the impacts of eddy diffusion as discussed in (e.g., Rambaldi and Mo, 1984; Legras and Ghil, 1985; Ferreira and Schubert, 1997). The governing equation (1) for the horizontal streamfunction has been extensively used in previous studies to examine the quasi-geostrophic dynamics under different large-scale conditions (e.g., Charney and DeVore, 1979; Legras and Ghil, 1985; Rambaldi and Mo, 1984; Schar, 1990).

To be specific for our TCG problem, we will apply Eq. (1) for a zonally periodic tropical channel, which is defined as

$$\Omega = [0, L_x] \times [0, L_y], \tag{2}$$

where $L_y$ is the width of the tropical channel in a hemisphere and $L_x$ is the zonal length of the channel. This domain roughly represents a region where the ITCZ can be treated as a long band wrapping around the Equator. For the current Earth's condition, $L_x \sim 40,000$ km, and $L_y \sim 1,000 - 1,500$ km (i.e., 10-15$^o$), and so by definition $L_x \gg L_y$.

---

[1]In Charney and DeVore (1979), the relaxation time $\alpha$ is proportional to the ratio of the Ekman depth $D_E$ over the depth of the fluid $H$, while the external forcing term $F$ can be treated as a large-scale vorticity source.





**Table 1.** Parameters of the model

| $Variable$ | $Range$ | $Remark$ |
|---|---|---|
| $U_o$ | 10-20 m s$^{-1}$ | Mean easterly flow in the tropical lower troposphere |
| $L_y$ | 1200-1500 km | Width of the tropical channel $\Omega$ |
| $L_x$ | $\sim$40,000 km | Length of the tropical channel domain $\Omega$ |
| $a$ | $\frac{2L_y}{L_x}$ | Aspect ratio of the tropical channel |
| $\alpha$ | $10^{-5} - 10^{-7}$s$^{-1}$ | Relaxation time |
| $\nu$ | $10 - 10^4$s$^{-1}$ | Horizontal eddy viscosity coefficient |
| $\beta$ | $2 \times 10^{-11}$s$^{-1}$ | Variation of the Coriolis parameter with latitudes |
| $\gamma$ | $10^{-10} - 10^{-11}$s$^{-2}$ | Magnitude of the external mass source/sink in the ITCZ breakdown model |

Before we can analyze the ITCZ breakdown model, it is necessary to have first an explicit expression for the forcing term $F$. In the early study by Ferreira and Schubert (1997), $F$ represents a mass sink that is a piecewise unit step function of latitudes. To account for the existence of the zonal jet in mid-latitude regions, Legras and Ghil (1985) however used a forcing of the form $F = \alpha \nabla \psi^*$, where $\psi^*$ is a given streamfunction that represents the zonal jet around $50^o N$. Given our focus on the ITCZ

dynamics, we will choose this forcing term such that its corresponding steady state can best represent the typical background flow in the tropical lower troposphere. A particular functional form for $F$ that meets this requirement is

$$F = \gamma \sin \frac{\pi y}{L_y} \tag{3}$$

where $\gamma$ denotes the strength of the forcing. Note that this forcing amplitude is not arbitrary, because its value dictates the zonal mean flow in the tropical region as will be shown below.

While the forcing term given by Eq. (3) differs from the unit step function in Ferreira and Schubert (1997), it turns out that (3) allows a steady solution consistent with the typical flow near the ITCZ. Indeed, the steady state $\psi_S$ of (1) resulted from this forcing is

$$\psi_S = \frac{-\gamma L_y^4}{\nu_e \pi^4 + \alpha L_y^2 \pi^2} \sin \frac{\pi y}{L_y} \tag{4}$$

The horizontal flow corresponding to this steady streamfunction is illustrated in Fig. 2, which shows two opposite easterly and

westerly flows to the north and the south of an ITCZ during a typical Northern Hemisphere summer as expected.

Given the above forcing $F$ and its corresponding steady state, the problem of the ITCZ breakdown is now mathematically reduced to the study of the instability of the steady-state (4) as the model parameters such as the forcing amplitude $\gamma$, the relaxation time $\alpha$, or the beta effect vary. To this end, it is more convenient to re-write Eq. (1) in the non-dimensional form such that our subsequent mathematical analyses can be simplified. Given the governing equation (1), it is apparent that the

natural scaling for time, streamfunction, and distance can be chosen respectively as follows:

$$t = \frac{1}{L\beta}t^*, \quad \psi = LU_0 \psi^*, \quad (x,y) = L(x^*,y^*), \quad F = \frac{\alpha U_0}{L}F^*,$$



where the asterisk denotes the nondimensional variables, and $U_0$ is a given characteristic horizontal velocity that determines the strength of the zonal mean flow in the tropical region. Nondimensionalizing Eq. (1) and neglecting the asterisks hereinafter, the nondimensional form for Eq. (1) becomes

$$\frac{\partial \Delta \psi}{\partial t} + \epsilon J(\psi, \Delta \psi) = E\Delta^2 \psi + F - A\Delta \psi - \frac{\partial \psi}{\partial x}, \tag{5}$$

where

$\epsilon = \dfrac{U_0}{L^2\beta}$  is the Rossby number,

$E = \dfrac{\nu_e}{L^3\beta}$  is the Ekman number,

$A = \dfrac{\alpha}{L\beta}$  is the ratio of the relaxation time to the inherent time related to the Earth's rotation rate.

For the sake of mathematical convenience, we will hereinafter extend the model domain from $[0, L_y]$ to $[-L_y, L_y]$ such that
the boundary conditions become meridionally symmetric along the Equator at $y = 0$. This mathematical method of extending the domain will simplify a lot of calculations, while has no effect on our solutions so long as we limit the final solution in the original domain $[0, L_y] \times [0, L - y]$ and maintain the Neumann boundary at $y = 0$ as shown below. The non-dimensional extended domain is therefore given by

$$\Omega = \left[0, \frac{2}{a}\right] \times [-1, 1].$$

where the scale factor $a \equiv 2L_y/L_x$ is introduced to simplify our spectral analyses. Given the above nondimensionlization, the non-dimensional form of the forcing (3) is now simply

$$F = \gamma_1 \sin \pi y, \text{ where } \quad \gamma_1 = \frac{\gamma L}{\alpha U_0}, \tag{6}$$

and the steady-state (4) is

$$\psi_S = -\frac{A\gamma_1}{E\pi^4 + A\pi^2} \sin \pi y. \tag{7}$$

We examine next the stability of the steady state (7) and how this critical point would bifurcate into new states as the model parameters vary, using the dynamical transition framework developed by Ma and Wang (2013). To this goal, it is necessary to study the behaviors of a deviation $\psi'$ around the given equilibrium (4). We follow the standard procedure in the dynamical transition and expand the solution around the critical point (5) in the form $\psi = \psi_S + \psi'$. It should be particularly emphasized here that unlike the traditional linear stability analyses in which one often assumes $\psi' \ll \psi_S$, the dynamical
transition framework does not impose such a condition on $\psi'$. The sole purpose of introducing the partition $\psi = \psi_S + \psi'$ is to simply shift the location of the stability analyses towards the steady state $\psi_S$, much like shifting the coordinate origin from 0 to a new critical point in any linear stability analyses. In Ma and Wang (2013)'s dynamical transition framework, the full nonliterary is maintained such that possible analyses of the central manifold can be carried out, and so no assumption of



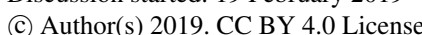


$\psi' \ll \psi_S$ is needed in our analyses here. With this partition, the corresponding governing equation for the perturbation $\psi'$ then becomes

$$\frac{\partial \Delta \psi}{\partial t} + \epsilon J(\psi, \Delta \psi) = E\Delta^2 \psi - A\Delta \psi - \frac{\partial \psi}{\partial x} + R\frac{d\widetilde{\psi_S}}{dy}\frac{\partial \Delta \psi}{\partial x} - R\frac{d^3\widetilde{\psi_S}}{dy^3}\frac{\partial \psi}{\partial x}, \tag{8}$$

where all the primes are hereinafter omitted for the sake of convenience, and a non-dimensional number $R$ and $\widetilde{\psi_S}$ are defined

as follows

$$R = \frac{\gamma_1 \epsilon}{E\pi^3 + A\pi}, \qquad \widetilde{\psi_S} = -\frac{\sin \pi y}{\pi}. \tag{9}$$

Given the nature of the ITCZ model, the periodic boundary conditions will be imposed in the zonal direction, and the free boundary conditions in the meridional direction for the perturbation equation(8) are applied at $y = -1$ and $y = 1$ such that

$$\psi(t, 0, y) = \psi\left(t, \frac{2}{a}, y\right),$$
$$\psi(t, x, -1) = \psi(t, x, 1) = 0, \tag{10}$$
$$\frac{\partial^2 \psi}{\partial y^2}(t, x, -1) = \frac{\partial^2 \psi}{\partial y^2}(t, x, 1) = 0.$$

The periodic boundary conditions along the west-east direction are naturally expected because of the cyclic property of the tropical channel around the Equator, while the free boundary conditions along the south-north direction will ensure that there is no meridional exchange (i.e., no $u$-wind component) at $y = -1$ and $y = 1$. Apparently, the Neumann boundary condition at the $y = 0$ is still valid after the domain extension, because of the continuity of the solution at $y = 0$ in the interior region.

To further reduce the governing equation of the perturbation as given by Eq. (8) in the familiar form that will facilitate our

analyses, we rewrite Eq. (8) in terms of an abstract functional notation that is standard in the study of the nonlinear dynamical transition. Introduce three differential operators $\mathcal{L}$, $\mathcal{G}$, and $\mathcal{A}$ as follows.

$$\mathcal{A}\psi \equiv \Delta\psi, \tag{11}$$

$$\mathcal{L}\psi \equiv E\Delta^2 \psi - A\Delta \psi - \frac{\partial \psi}{\partial x} + R\frac{d\widetilde{\psi_S}}{dy}\frac{\partial \Delta \psi}{\partial x} - R\frac{d^3\widetilde{\psi_S}}{dy^3}\frac{\partial \psi}{\partial x}, \tag{12}$$

$$\mathcal{G}\psi \equiv \epsilon J(\Delta\psi, \psi). \tag{13}$$

Eq. (8) for the perturbation streamfunction with boundary condition (10) can be then put into the following abstract operator form

$$\frac{d\mathcal{A}\psi}{dt} = \mathcal{L}\psi + \mathcal{G}(\psi). \tag{14}$$

As seen in this abstract form, the operators $\mathcal{A}$ and $\mathcal{L}$ are linear, whereas $\mathcal{G}$ is a nonlinear operator due to the Jacobian's term. A standard procedure to Eq. (14) is to employ the traditional bifurcation analyses and examine first the spectra of the eigenvalues

and eigenvectors of the linear component $\mathcal{L}$. We then determine the stability characteristics of the linear system, and finally



construct the central manifold function with the full nonlinear terms included so that the stable and/or unstable properties of the new states of Eq. (8) can be quantified as the model parameters vary. The outcomes from these analyses are i) the conditions on the large-scale environment that could determine the stability of the steady state as well as the upper bound on the number of tropical unstable disturbances, and ii) the structure of new states after the dynamical transition that the large-scale flows must

possess to allow for the formation of initial tropical disturbances. These outcomes are interesting, because they could allow us to quantify the maximum number of environmental tropical embryos that the ITCZ can support in the tropical channel, thus addressing the question of how many TCs that we would most expect in the tropical region at any given time.

## 3  An upper bound on unstable modes

### 3.1  Eigenmode analyses

We start first with the search for the entire spectrum of the eigenvalues $\rho$ of the linear operator $\mathcal{L}$ in (14). Define a linear operator $L(\rho)$ as follows

$$L(\rho)\psi = \mathcal{L}\psi - \rho\psi, \ \ \rho \in \mathbb{C}, \tag{15}$$

Then, all eigenvectors of the linear operator $\mathcal{L}$ are non-trivial solutions of $L(\rho)\psi = 0$ with a corresponding eigenvalue $\rho$. Because of the periodic boundary condition in the $x$-direction, it turns out that the eivenvectors cannot be arbitrary. Indeed, the

boundary conditions (10) impose a strict constraint on the possible functional forms of $\psi$ such that every eigenvectors $\psi$ of $\mathcal{L}$ must be expressed in the following separation form

$$\psi_m(x,y) = e^{i\pi max}\Psi(y), \tag{16}$$

where $m \in \mathbb{Z}$ is any integer, and $\Psi(y)$ is the perturbation amplitude. Denote the corresponding eigenvalue $\rho_m$ for each meridional mode $m$, a substitution of the preceding separation form into the eigenvalue equation $L(\rho_m)\psi = 0$ yields

$$\begin{cases} \mathcal{L}_m\Psi = \rho_m\mathcal{A}_m\Psi, \\ \Psi(-1) = \Psi(1) = \Psi''(-1) = \Psi''(1) = 0, \end{cases} \tag{17}$$

where the primes in Eq. (17) hereinafter denote the derivative of the streamfunction with respect to $y$, and the following new notations have been introduced:

$$\begin{aligned} \mathcal{L}_m\Psi \equiv &E(D^2 - a^2m^2\pi^2)^2\Psi - A(D^2 - a^2m^2\pi^2)\Psi - ima\pi\Psi \\ &+ ima\pi R\widetilde{\psi}'_S(D^2 - a^2m^2\pi^2)\Psi - ima\pi R\widetilde{\psi}'''_S\Psi, \end{aligned} \tag{18}$$

$$\mathcal{A}_m\Psi = (D^2 - a^2m^2\pi^2)\Psi, \text{ and} \tag{19}$$

$$D \equiv d/dy. \tag{20}$$

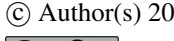



Applying the boundary conditions $\Psi(-1) = \Psi(1) = \Psi''(-1) = \Psi''(1) = 0$ to Eq. (17), it is immediate to see that all even-order derivatives of the perturbation amplitude $\Psi(y)$ vanish at the boundaries, i.e.,

$$\Psi^{(2n)}(-1) = \Psi^{(2n)}(1) = 0, \quad n = 0, 1, \cdots. \tag{21}$$

This important property of the perturbation amplitude $\Psi(y)$ results in a constraint that $\Psi(y)$ must be expressed in the following form

$$\Psi(y) = \sum_{n \geq 0} \phi_n \cos\left(n + \frac{1}{2}\right)\pi y + \sum_{n \geq 1} \widetilde{\phi}_n \sin n\pi y, \tag{22}$$

where $\phi_n$ and $\widetilde{\phi}_n$ are the coefficients to be determined by the eigenvalue equation. As a result, every solution $\psi_m(x,y)$ of $L(\rho)\psi = 0$ can be expressed as

$$\psi_m(x,y) = \sum_{n \geq 0} i^n e^{ima\pi x} \phi_{m,n} \cos\left(n + \frac{1}{2}\right)\pi y + \sum_{n \geq 1} i^n e^{ima\pi x} \widetilde{\phi}_{m,n} \sin n\pi y, \quad m \in \mathbb{Z}, \tag{23}$$

where we have re-defined the expansion coefficients as $i^n \phi_{m,n}$ and $i^n \widetilde{\phi}_n$ instead of $\phi_{m,n}$ and $\widetilde{\phi}_n$ as in (22) for the sake of convenience.

In what follows, we will determine the wavenumber $m$ such that the eigenvector $\psi_m$ given by (23) becomes first unstable, i.e., the real part of the corresponding eigenvalue $\rho_m$ turns to be positive, as the control parameter $R$ increases. It can be readily verified that for any complex eigenvalue $\rho_m \in \mathbb{C}$, $\psi_m$ and $L(\rho)\psi_m$ will have the same functional form. Thus, let us denote

$$L(\rho_m)\psi_m = L(\rho_m) \sum_{n \geq 0} i^n e^{ima\pi x} \phi_{m,n} \cos\left(n + \frac{1}{2}\right)\pi y + L(\rho_m) \sum_{n \geq 1} i^n e^{ima\pi x} \widetilde{\phi}_{m,n} \sin n\pi y \tag{24}$$

$$\equiv \sum_{n \geq 0} i^n e^{ima\pi x} \varphi_{m,n} \cos\left(n + \frac{1}{2}\right)\pi y + \sum_{n \geq 1} i^n e^{ima\pi x} \widetilde{\varphi}_{m,n} \sin n\pi y = 0. \tag{25}$$

Apparently, (23) is an eigenvector of the eigenvalue equation $L(\rho_m)\psi = 0$ if and only if the above identity is true $\forall (x,y)$. As a result, explicit calculation of each term in Eq. (24) will lead to

$$\varphi_{m,n} = B_{m,n+1}\phi_{m,n+1} + C_{m,n}\phi_{m,n} - B_{m,n-1}\phi_{m,n-1} = 0, \ n \geq 1, \tag{26}$$

$$\varphi_{m,0} = B_{m,1}\phi_{m,1} + C_{m,0}\phi_{m,0} + i\left(A_{m,0}\phi_{m,0} - \phi_{m,0}\right) = 0, \ n = 0, \tag{27}$$

$$\widetilde{\varphi}_{m,n} = D_{m,n+1}\widetilde{\phi}_{m,n+1} + E_{m,n}\widetilde{\phi}_{m,n} - D_{m,n-1}\widetilde{\phi}_{m,n-1} = 0, \ n \geq 2 \tag{28}$$

$$\widetilde{\varphi}_{m,1} = D_{m,2}\widetilde{\phi}_{m,2} + E_{m,1}\widetilde{\phi}_{m,1} = 0, \ n = 1, \tag{29}$$



where the coefficients $A_{m,n}, B_{m,n}, C_{m,n}, D_{m,n}, E_{m,n}$ are

$$
\begin{cases}
A_{m,n} = a^2 m^2 + (n+1/2)^2 \\
B_{m,n+1} = (1 - A_{m,n+1}) \\
C_{m,n} = \frac{2\pi^3 E A_{m,n}^2 + 2\pi(A+\rho_m)A_{m,n} - 2iam}{am\pi^2 R}
\end{cases}
\quad , \quad n \geq 0, \quad |m| \geq 1,
\tag{30}
$$

$$
\begin{cases}
B_{0,n+1} = (1 - A_{0,n+1}) \quad A_{0,n} = (n+1/2)^2 \\
C_{0,n} = 2\pi^3 E A_{0,n}^2 + 2\pi(A+\rho_m)A_{0,n}
\end{cases}
\quad , \quad n \geq 0, \quad m = 0,
$$

$$
\begin{cases}
D_{m,n} = (1 - \widetilde{A}_{m,n}), \quad \widetilde{A}_{m,n} = a^2 m^2 + n^2 \\
E_{m,n} = \frac{E\pi^3 \widetilde{A}_{m,n}^2 + \pi(A+\rho_m)\widetilde{A}_{m,n} - i2am}{am\pi^2 R}
\end{cases}
\quad , \quad n \geq 1, \quad |m| \geq 1,
\tag{31}
$$

$$
\begin{cases}
D_{0,n} = (1 - \widetilde{A}_{0,n}), \quad \widetilde{A}_{0,n} = n^2 \\
E_{0,n} = E\pi^3 \widetilde{A}_{0,n}^2 + \pi(A+\rho_m)\widetilde{A}_{0,n}
\end{cases}
\quad , \quad n \geq 1, \quad m = 0.
$$

Given the conditions (26)-(29), a simple way to obtain the amplitudes $\phi_{m,n}$ and $\widetilde{\phi}_{m,n}$ is to group all coefficients $\phi_{m,n}$ in each
of these identities. This can be done effectively by multiplying the conjugate coefficient $\overline{\phi_{m,n}}$ and a factor $B_{m,n}$ on both sides
of (26)-(27), and similarly $\overline{\widetilde{\phi}_{m,n}}$ and a factor $D_{m,n}$ on both sides of (28)-(29). Adding the resulting identities together and
taking the sum over $n$ allows us to extract a relationship between the amplitudes of $\phi_{m,n}$ and the eigenvalue $\rho_m$ as follows:

$$
\sum_{n \geq 0} B_{m,n} \varphi_{m,n} \overline{\phi_{m,n}} = 0,
\tag{32}
$$

$$
\sum_{n \geq 1} D_{m,n} \widetilde{\varphi}_{m,n} \overline{\widetilde{\phi}_{m,n}} = 0.
\tag{33}
$$

Note that all pairs of the form $(B_{m,n+1} B_{m,n} \overline{\phi}_{m,n+1} \phi_{m,n}, B_{m,n+1} B_{m,n} \phi_{m,n+1} \overline{\phi}_{m,n})$ in Eq. (32) are conjugated to each other
so that their sum will produce a purely imaginary number. As a result, the real parts of (32) and (33) must come from the term
involving $C_{m,n}$ and can be therefore obtained as

$$
\sum_{n \geq 0} B_{m,n} A_{m,n} (E\pi^2 A_{m,n} + A + \Re[\rho_m]) |\phi_{m,n}|^2 = 0,
\tag{34}
$$

$$
\sum_{n \geq 1} D_{m,n} \widetilde{A}_{m,n} (E\pi^2 \widetilde{A}_{m,n} + A + \Re[\rho_m]) |\widetilde{\phi}_{m,n}|^2 = 0.
\tag{35}
$$

By imposing the physical requirement on the existence of the eigenmodes with $\phi_{m,n} \neq 0$ and $\widetilde{\phi}_{m,n} \neq 0$, Eqs. (34)-(35) can
provide a great insight into the stability and structure of the eigenmodes that we will now turn into.

## 3.2 Upper bound on the unstable eigenmode

Eqs. (34)-(35) contain a number of very powerful properties. First, one notices that the real part of the eigenvalue $\rho_m$ for
$m = 0$, if exist, must be negative due to the properties that the coefficients $A > 0$, $A_{m,n} > 0$, $B_{0,n} \leq 0$ and $D_{0,n} \leq 0$. Indeed,



if we assume that there exists an eigenvector $\psi_m$ such that $\Re[\rho_m] > 0$ for $m = 0$, then it can be directly seen from the quadratic form of (34) that

$$\phi_{m,n} = 0, \ \widetilde{\phi}_{m,n} = 0, \qquad n \geq 0,$$

and so there would exist no solution at all, which contradicts our assumption of the existence of the eigenvector for $m = 0$.
Thus, the model $m = 0$ is always stable. Because this stable mode does not allow us to examine any transition behaviors, this special mode will not be considered hereafter.

For $m \neq 0$, it can be seen also from (34) that all possible unstable eigenvectors with $m \neq 0$ must satisfy the following constraints

$$\begin{cases} \begin{cases} \phi_{m,n} = 0, \ \text{when,} \ a \geq \frac{\sqrt{3}}{2}, \ m \in Z; \\ \phi_{m,n} = 0, \ \text{when,} \ \frac{\sqrt{3}}{4} \leq a < \frac{\sqrt{3}}{2}, \ |m| \geq 2, \\ \phi_{m,n} = 0, \ \text{when,} \ \frac{\sqrt{3}}{6} \leq a < \frac{\sqrt{3}}{4}, \ |m| \geq 3, \\ \dots \quad \vdots \quad \dots \\ \phi_{m,n} = 0, \ \text{when,} \ \frac{\sqrt{3}}{2k} \leq a < \frac{\sqrt{3}}{2k-2}, \ |m| \geq k \end{cases} \\ \widetilde{\phi}_{m,n} = 0, \ n \geq 1, \ \text{for all} \quad a > 0. \end{cases} \qquad (36)$$

This conclusion can be explicitly proven if we note again that the constraint (36) will ensure that the coefficient $A_{m,n} > 0$, and $B_{m,n} < 0$. If we assume that there exists any unstable eigenvector $\psi_m$ with some wavenumber $m \neq 0$ such that the corresponding eigenvalue $\rho_m$ satisfies $\Re[\rho_m] > 0$, then Eq. (34) immediately indicates that $\phi_{m,n} = 0, \forall \ |m| \geq k$ and $n \in \mathbb{Z}^+ \cup \{0\}$ (i.e., $\psi_m = 0$), and so no such unstable eigenvector $\psi_m$ can exist at all. As a result, we obtain a remarkable result that any possible unstable modes must be constrained by the condition $|m| \leq k$, where $k$ is an integer satisfying $\frac{\sqrt{3}}{2k} \leq a < \frac{\sqrt{3}}{2k-2}$.

To help understand the significance of this result, we consider a tropical channel domain between $10^o$S-$10^o$N in the Earth's atmosphere (i.e., $L_y \sim 1200$ km) and $L_x \sim 40,000$ km such that $a \equiv 2L_y/L_x \approx 0.06$. Using the condition $\frac{\sqrt{3}}{2k} \leq a < \frac{\sqrt{3}}{2k-2}$, we obtain an upper bound wavenumber $k \approx 12$. Thus, the largest integer number $m$ for this given tropical channel domain is upper bounded. Although we do not know exactly in advance which wavenumber $m < k$ will become unstable, because the condition $|m| < k$ includes a range of $m$ whose real part $\Re[\rho_m]$ could be positive, the above result is still very significant due to
its explicit indication that the unstable wavenumbers cannot be arbitrary but must be bounded. Any eigenvectors with $|m| \geq k$ must be therefore stable and cannot grow.

A natural consequence of the above result is that not only the total number of TC disturbances is bounded from above, but also the size of these disturbances must be limited as well (i.e., the size of each disturbance is $\sim L_x/m$). If we assume that each of these disturbances could be eventually responsible for one TC embryo, the upper limit in the number of the disturbances as
found from the above condition would imply a lower bound on the overall size of TCs, which has to be larger than $3 \times 3000$ km in diameter. That is, the TC size on the Earth's atmosphere cannot be arbitrarily small, but must be larger than a limit of $\sim 10^3$ km, a fact that has been long observed but not fully explained so far.





It should be stressed that, the condition on the unstable modes derived from the eigenvalue $\Re[\rho_m]$ as seen from (36) is just a necessary condition, and it is by no means sufficient to specifically know which wavenumber in the range of $[1, k]$ will become first unstable. Thus, our next task is to examine how the real part of the eigenvalue $\Re[\rho_m]$ varies as the model parameter $R$ increases for each value of $m$. Note that the non-dimensional number $R$ encodes several important large-scale conditions

including the Rossby number, the Ekman number, and the strength of the background flow $U_0$ as seen in (9). As these large-scale conditions change, $R$ will vary as well. Depending on how the eigenvalue $\rho_m$ varies as a function of $R$, there may emerge a positive eigenvalue $\Re[\rho_m]$ that we need to quantify.

To ensure the existence of such a positive eigenvalue as $R$ increases, it is necessary to show that $\Re[\rho_m]$ must be an increasing function of $R$ such that the real part can become positive as $R$ increases. The specific wavenumber $m$ for which $\Re[\rho_m]$ first

becomes positive will possess the structure that dictates the new dynamical transition of the system, according to the Principle of Exchange of Stabilities (see Appendix 1 for the definition of this Principle). Due to the complication in deriving the exact expression for $\rho_m$, details of the derivations of $\Re[\rho_m]$ as a function of $R$ are provided in Appendix 2. An important conclusion from these derivations is that $\lim_{R \to +\infty} \Re[\rho_m(R)] = +\infty$, which then implies that there indeed exists a critical value $R^*$ at which $\Re[\rho_m](R^*) = 0$. This requirement is critical, since it directly indicates that the Principle of Exchange of Stabilities is

ensured. More strictly speaking, this result means that there exists a positive integer $n \le k$ and a critical Reynolds number $R^* > 0$ such that the following conditions

$$
\begin{cases}
\Re[\rho_{n,1}] = \Re[\rho_{-n,1}] \begin{cases} > 0 & \text{if } R > R^*, \\ = 0 & \text{if } R = R^*, \ \forall n = m_1,, \cdots, m_l, \\ < 0 & \text{if } R < R^*, \end{cases} \\
\Re[\rho_{m,k}] < 0, \qquad \text{if } (m,k) \ne (m_i, 1), \quad 1 \le i \le l, \\
\Im[\rho_{n,1}(R)] \ne 0 \quad \text{for } R \ge R^*,
\end{cases}
\tag{37}
$$

must hold true. Corresponding to the first unstable mode $m$ and eigenvalue $\rho_{n,1}$, its eigenvector is then given by

$$
\psi_m = \sum_{n=0}^{\infty} i^n e^{ima\pi x} \phi_{m,n} \cos \left( n + \frac{1}{2} \right) \pi y, \ 1 \le |m| \le k.
$$

Note that these eigenvectors are unstable for $R > R^*$ and $|m| < k$ only, because all other eigenvectors ($|m| > k$) are always stable as shown by the condition (36).

Due to the complicated expression for the eigenvalue $\rho_m(R)$ as shown in Appendix 2, the value $R^*$ cannot be exactly derived but must be numerically approximated for each $m$. The proof of $\lim_{R \to +\infty} \Re[\rho_m(R)] = +\infty$ provided in Appendix 2 ensures that $R^*$ always exists, and so it should be found numerically. Fig. 3 shows the critical value $R_m^*$ as a function of $2/a$ for each

value of $m$, which is obtained by using a numerical approximation. Note that for each value of $a$, there will exist only one value $k$ that satisfies $\frac{\sqrt{3}}{2k} \le a < \frac{\sqrt{3}}{2k-2}$ and a value $m < k$ such that $\Re[\rho_{m,1}] = 0$. By searching for the value of $R_m^*$ that ensures $\Re[\rho_{m,1}] = 0$, we obtain for each $m \le k$ a curve $R_m^* = R_m^*(a)$ that determines the onset of the dynamical transition. Because the eigenvalues and the eigenfunctions corresponding to $-m$ are the complex conjugate of the respective eigenvalues and the eigenfunctions corresponding to $m$, only the cases of nonnegative $m$ need to be examined.





As shown in Figure 3, there are several key differences between the asymptotic limits of a very small and a very large $a$. Specifically, for a larger value of $a$ (i.e., a wider tropical region), the smaller wavenumbers $m$ will become unstable first, starting with $m = 5$, which then decreases for a larger $R$. For the smaller value of $a$ (i.e., a narrower tropical channel), the larger wavenumbers will however become unstable first as shown in Figure 3. For example, for the typical scales of the Earth's tropical region with $L_x \approx 40,000$ km, and $L_y \approx 1,200$ km, $2/a = L_x/|L_y| \approx 33.3$. According to Fig. 3, the wavenumber $m = 9$ will become unstable first as $R$ crosses the value $R^* = 4$. Thus, the dynamical transition for $m = 9$ will produce a new unstable wave structure corresponding to $m = 9$ at the bifurcation point. As the parameter $R$ increases, other unstable modes corresponding to $m = 8, 7, 6...$ start to emerge, thus producing more unstable structure as a result of the dynamical transition.

To focus on the wavenumber that is first unstable instead of the critical number $R^*$ as shown in Figure 3, Figure 4 shows the first unstable mode $m$ as a function of $a$, assuming all same parameters used in Figure 3. It can be seen in this Figure 4 that for each value of $a$, there is only one wavenumber $n = n(a)$ for which $R_n^* = \min_{m \in \mathbb{N}} R_m^*$. This is the critical value $R^* = R_n^*$ at which the dynamical transition will occur according to the PES condition. In addition, one can better see how the first unstable mode depends on the aspect ratio of the tropical channel, with a higher wavenumber for a narrower tropical region. This same behavior is also valid for a range of values of the Ekman number $E$ and Rossby number $\epsilon$, which is not shown here because they do not provide any further information.

## 4 Bifurcation structure

While the analyses in the previous section could show an upper bound on possible unstable modes, the structure of the unstable modes as well as the subsequent effects of the nonlinear terms have not been discussed. Depending on the eigenvalues and the structure of the eigenvectors when the first dynamical transition takes place, one can reduce the full nonlinear system (14) to a central manifold and construct a central manifold function to examine the bifurcation and the structure of the new state with all nonlinear terms included. Standard procedure in dynamical transition Ma and Wang (2013) shows that once the steady-state $\psi_S$ loses its stability for $R > R^*$, the supercritical Hopf bifurcation may occur with a new stable state approximated as follows

$$\psi = \psi_S + \left( \frac{\Re(\rho_{n,1})}{|\Re(A)|} \right)^{\frac{1}{2}} f_n(x, y, t) + h.o.t. \tag{38}$$

assuming that the nondimensional parameter $R$ is sufficiently close to $R^*$, i.e.,

$$0 < \frac{R - R^*}{R^*} \ll 1.$$

Using a higher-order approximation around the critical point on the extended central manifold, it can be shown that the manifold function could provide a better approximation for $\psi$ when $R > R^*$ (Kieu et al. 2018). Nonetheless, the smooth behaviors of the eigenvector at $R = R^*$ for the supercritical Hopf bifurcation suffices to indicate that the structure of the solution at $R = R^*$ can well represent the behavior of the new stable solution near $R = R^*$. Note that there may appear either Hopf bifurcation or double Hopf bifurcation, depending on the transition multiplicity at the critical value. This subtlety will introduce much more complex analysis of the bifurcation and a transversal intersection in the parameter plane that we will not present herein.




While these higher-order derivations of the central manifold function require some technical details that are beyond the scope of this study (see Kieu et al. 2018), it is possible to approach the bifurcation structure problem by numerically solving the eigenvalue problem (18). Specifically, we notice that the $x$ dependence can be obtained by simply searching for the first unstable mode $m$ as $R$ approaches the critical value $R^*$. Using this numerical approach to find the critical value of $R^*$, the

entire spectrum of eigenvectors associated with the potential new stable states after the dynamical transition can be found for each set of large-scale environmental parameters. We note at this point that the exact mode $m$ at which the eigenvector becomes unstable is dependent on $R$ as shown in Figures 3 and 4. The only constraint we are certain is that $|m| < k$. Thus, the new stable mode for $R > R^*$ could inherit the structure of any value of $|m| < k$ at $R = R^*$. This numerical approach is powerful, as it allows one to search for not only the critical parameter $R^*$ at which the PES condition is ensured, but also the structure of

new stable states for any value of $R > R^*$ after the bifurcation point.

To illustrate the results from this numerical approach, we assume the following set of the large-scale environmental conditions in the typical tropical region

$$L_y \sim 1000\text{km}, \quad U_0 \sim 10\text{m s}^{-1}, \quad \alpha \sim 10^{-6}\text{s}^{-1}, \quad \beta \sim 10^{-11}\text{m}^{-1}\text{s}^{-1}, \quad \nu = 1000\text{m}^2\text{s}^{-1},$$

which result in a Rossby number $\epsilon \approx 0.5$ and an Ekman number $E \approx 0.05$. Further use of Eq. (4) for the steady state and note

that $U_0 = \partial \psi_S / \partial y$, one then obtains also an estimation for the forcing amplitude $\gamma \approx 7 \times 10^{-10} s^{-2}$. From the definition of the nondimenisonal number $R$, we then get $R \approx 4.8$, which is above the critical value $R^* \approx 4$ for $m = 9$ as shown in Figure 3. Thus, the PES condition is met, and a new stable structure must emerge after the dynamical transition. As a result, the eigenvalue problem (17) needs to be solved for the first eigenvector and its dual eigenvector, given this value for $R$. For this numerical method, we use a Legendre-Galerkin method where the unknown fields are expanded using a basis of $N$ polynomials, which

are compact combinations of the Legendre polynomials satisfying the four boundary conditions (17) (Shen et al., 2011) for the details of this numerical method). For the convergence of the numerical scheme, $N = 100$ is sufficient. Once the eigenvector problem is solved, a further approximation on the central manifold can be applied so that we can examine the stability of different states around the critical point on the central manifold.

Figure 5 shows a new state as a result of the dynamical transition for $R > R^*$, which is obtained by from the numerical

procedures described above. Among several significant features of this numerical solution, the first noteworthy one is that the new state possesses a large-scale structure consistent with the ITCZ breakdown as observed in the idealized simulations by Ferreira and Schubert (1997). Specifically, the tropical channel contains 10 large-scale disturbances, each has the horizontal scale of about 5000 km that could serve as embryos for the subsequent TC formation. Furthermore, this new state moves to the left with a period of $T \sim 3.2116$ (i.e., $T \approx 4$ days in the physical dimensional unit) as a result of nonzero imaginary part of the

eigenvalues. It should be mentioned that the results shown in Figure 5 is hold for the Earth's tropical channel with $L_x/L_y \approx 36$ (or equivalently $a \approx 1/17$). For a different domain configuration such as different planets or climate with different tropical width, the unstable mode may be different and the maximum number of the TC-favorable disturbances will change as well.

That these large-scale structure of disturbances moving to the left with the temporal scale of $\sim 4$ days as a consequence of the dynamical transition shown above is quite remarkable, because these westward moving disturbances are to some extent




similar to easterly waves in the real atmosphere. In this regard, the easterly waves that are often associated with the TCG can be now seen as a natural consequence of the dynamical transition process, even for barotropic flows. Such consistency between the observed and theoretical estimation of the large-scale modes in the tropical region indicates that the barotropic instability and its inherent nonlinear dynamics can account for the pre-conditioning environment for TC genesis.

We should emphasize that the large-scale structure shown in Figure 5 does not itself dictate that the disturbances have to grow and turn into TCs. Instead, these structures are simply new stable periodic solutions after the dynamical transition occurs. That is, for $R < R^*$, the stable structure is the steady state as given in Figure 2, whereas the new stable structure shown in Figure 5 will emerge after $R > R^*$. As soon as these stable structures emerge, the subsequent dynamic-thermodynamic feedback may be triggered, which result in further growth of the disturbances within each wave, similar to the pouch model proposed in

Dunkerton et al. (2012). The subsequent rapid intensification of the tropical disturbances requires various detailed physics that are, however, not the focus of this work, and so will not be hereinafter discussed. In this regard, the new stable periodic state shown in Figure 5 serves only as a pre-conditioning environment for incipient disturbances to grow.

## 5   Conclusion

In this study, we examined the dynamical mechanisms underlying the large-scale formation of tropical cyclones (TCs), using

the barotropic model for the Intertropical Convergence Zone (ITCZ) driven by an external mass forcing. Assuming a variant type of a forcing that mimics the mass sink/source in the ITCZ as previously used in Ferreira and Schubert (1997), it was proved that the large-scale steady flow (i.e., the critical point) in the ITCZ model loses its stability for a bounded range of the wavenumber $|m| < k$ if large-scale environmental conditions including the magnitude of the mean flow, the Ekman number, and the Rossby number satisfy a certain constraint. That the number of the TC disturbances in the tropical region is upper

bounded in any given day could alone offer an explanation for a fundamental question of why the Earth's atmosphere can support a limited number of TCs globally each year.

Using the Principle of Exchange of Stabilities condition for the ITCZ model, we found that the model undergoes a bifurcation and associated dynamical transition, which helps further determine the maximum number of TC disturbances that the Earth's atmosphere can generate Specifically, the theoretical estimation of the largest wavenumber $k$ that can still support the unstable

structure as a result of the ITCZ breakdown is $k \sim 12$, assuming the typical characteristic of the Earth's tropical channel in which the zonal scale of the tropical channel is about order of magnitude larger than the width of the channel. This constraint on the largest wavenumber of the unstable eigenmodes imposes not only an upper limit on the number of TC disturbances in the tropical region, but also results in a lower bound on the size of TC disturbances. This lower bound on the size of the tropical disturbances may help explain why TCs cannot be arbitrary small, but must be larger than a certain limit.

To verify our theoretical analyses, a numerical method is used to search for the structure on the central manifold of the ITCZ model as the model parameter $R$ is increased to a value larger than a critical value $R^*$. Here, the key parameter $R$ controlling



the bifurcation in our ITCZ model is given by

$$R = \frac{\gamma \epsilon \pi}{E\pi^4 + A\pi^2},$$

where $\gamma$ is parameter measuring the strength of the ITCZ mass sink/source, $A$ is the parameter measure the effect of surface drag, $\epsilon$ is a parameter measuring the mean zonal flow, and $E$ is the Ekman number representing the eddy viscosity. Our numerical results confirmed that for $R > R^*$, a new large-scale state emerges whose structure depends on the value $R$. For $R$ sufficiently close to the critical value $R^*$, the new state possesses a new type of periodic motion with two groups of symmetric disturbances across the Equator. These new stable periodic solutions describe a type of westward-moving disturbances within the ITCZ, very similar to the classical easterly waves in the tropical region. The findings obtained from the ITCZ breakdown model in this study thus provide a new insight into the formation of TC disturbances in the Earth's tropical atmosphere, and provide a rigorous proof for the observation of the limited number of TCs annually at the global scale.

## Appendix A: Principle of Exchange of Stabilities

The Principle of Exchange of Stabilities (PES) for a dynamical system basically refers to a critical condition for which the eigenvalues of the linear operator first cross a prescribed value. More precisely, the PES can be precisely stated as follows.

Let $\mathbf{L}_\lambda$ and $\mathbf{G}$ represent the linear and nonlinear parts of a dynamical system in the abstract form:

$$\frac{d\mathbf{u}}{dt} = \mathbf{L}_\lambda(\mathbf{u}) + \mathbf{G}(\mathbf{u}, \lambda) \tag{A1}$$

where $\lambda \in \mathbb{R}$ is the model parameter, and $\mathbf{u} \in \mathbb{R}^n$ represents the state of the system. By defniition, $\mathbf{L}_\lambda$ is a parameterized linear operator that depends continuously on $\lambda$. Consider the eigenvalue equation given by

$$\mathbf{L}_\lambda \mathbf{e} = \beta(\lambda)\mathbf{e}, \tag{A2}$$

where $\mathbf{e}$ is eigenvector, and $\beta(\lambda) \in \mathbb{C}$ the eigenvalue. Let $\beta_j(\lambda) \in \mathbb{C} | j \in \mathbb{N}$ be the eigenvalues (counting multiplicity) of $\mathbf{L}_\lambda$. If we have

$$\begin{cases} \Re[\beta_j(\lambda)] \begin{cases} < 0 & \text{if } \lambda < \lambda_0, \\ = 0 & \text{if } \lambda = \lambda_0, \; \forall 1 \le i \le m \;, \\ > 0 & \text{if } \lambda < \lambda_0, \end{cases} \\ \Re[\beta_j(\lambda_0)] < 0, \; \forall j \ge m+1, \end{cases} \tag{A3}$$

then the system is said to satisfy the PES condition at $\lambda_0$, which signifies a bifurcation of the system from one state to another. For dissipative systems, the PES condition has a much more powerful implication than a simple bifurcation, as it ensures a dynamical transition that can be completely categorized by three different types of transition including the continuous transition, the catastrophic transition, and the random transition. See Ma and Wang (2013) for more details of the PES conditions for nonlinear systems.





## Appendix B: Existence of the critical number $R^*$

For $\frac{\sqrt{3}}{2k+2} \leq a < \frac{\sqrt{3}}{2k}$ and $1 \leq m \leq k$ $(k = 1, 2, \cdots)$, it is easy to see from (34) that we must have

$$\phi_{m,0} \neq 0,$$

because otherwise we will have $\phi_{m,n} = 0, n \geq 0$, and there would exist no eigenvectors. For the sake of convenience, we will

hereinafter replace $\psi_m$ by $\frac{\psi_m}{B_{m,0}\phi_{m,0}}$, and similarly replace $\phi_{m,n}$ by $\frac{\phi_{m,n}}{B_{m,0}\phi_{m,0}}$. Equation (**??**) is then re-written as follows:

$$B_{m,n+1}\phi_{m,n+1} + C_{m,n}\phi_{m,n} - B_{m,n-1}\phi_{m,n-1} = 0, \ \ n \geq 1,$$
$$B_{m,1}\phi_{m,1} + C_{m,0}\phi_{m,0} + i\left(A_{m,0}\phi_{m,0} - \phi_{m,0}\right) = 0, \ \ n = 0, \tag{B1}$$

and

$$\sum_{n\geq 0} B_{m,n}A_{m,n}(E\pi^2 A_{m,n} + A + \Re[\rho_m])|\phi_{m,n}|^2 = 0. \tag{B2}$$

Denote

$$d_{m,n} = \frac{C_{m,n}}{B_{m,n}},$$

and let

$$\eta_{m,n} = B_{m,n}\phi_{m,n},$$

(B1) can be further rewritten as

$$\eta_{m,n+1} + d_{m,n}\eta_{m,n} - \eta_{m,n-1} = 0, \quad n \geq 1,$$
$$\eta_{m,1} + d_{m,0} - i = 0, \quad n \geq 0. \tag{B3}$$

This reduced equation (B3) allows us to deduce a number of important constraints. Indeed, we re-arrange (B3) as follows:

$$-d_{m,0} + i = \eta_{m,1}, \quad \eta_{m,0} = 1,$$
$$\xi_{m,n} = \frac{\eta_{m,n}}{\eta_{m,n-1}} = \frac{1}{d_{m,n} + \frac{\eta_{m,n+1}}{\eta_{m,n}}},$$
$$-d_{m,0} + i = \frac{1}{d_{m,1} + \xi_{m,2}} = \frac{1}{d_{m,1} + \frac{1}{d_{m,2}+\xi_{m,3}}}. \tag{B4}$$

It is readily seen from (B3) that $\eta_{m,n} = 0$ for all $n \geq 0$ whenever there exists a $l \geq 0$ for which $\eta_{m,l} = 0$. This means that $\xi_{m,n} \neq 0$ for all $n \geq 0$. From (B4) one can derive that

$$\eta_{m,n} \equiv \xi_{m,1}\xi_{m,2}\cdots\xi_{m,n}, \ n \geq 1. \tag{B5}$$





Therefore, for $\frac{\sqrt{3}}{2k+1} \le a < \frac{\sqrt{3}}{2k}$ $(k=1,3,3,...)$, (B2) can be equally rewritten as:

$$
\begin{cases}
\sum_{n \ge 0} \Re[d_{m,n}]|\eta_{m,n}|^2 = 0, \\
\Re[d_{m,n}] < 0 (n \ge 1), \Re[d_{m,0}] > 0,
\end{cases} \qquad m \le k.
\tag{B6}
$$

One can deduce from the third equality of (B4) that

$$
\rho_m = -A - \pi^2 E A_{m,0} + \frac{2iam + iam\pi^2 R(1 - A_{m,0})}{2\pi A_{m,0}} + \frac{\frac{-am\pi R(1-A_{m,0})}{2A_{m,0}}}{d_1 + \xi_{m,2}}.
\tag{B7}
$$

Let's define a function $F$ using right hand side of (B7), i.e.,

$$
F(\rho_m, R) = -A - \pi^2 E A_{m,0} + \frac{2iam + iam\pi^2 R(1 - A_{m,0})}{2\pi A_{m,0}} + \frac{\frac{-am\pi R(1-A_{m,0})}{2A_{m,0}}}{d_1 + \xi_{m,2}}.
$$

Due to the fact that

$$
\Re d_{m,n} < 0 (n \ge 1),
$$

we can obtain that

$$
\quad |F(\rho_m, R)| \le \left| -A - \pi^2 E A_{m,0} + \frac{2iam + iam\pi^3 R(1 - A_{m,0})}{2\pi A_{m,0}} \right|
$$

$$
+ \frac{\left| \frac{-am\pi R(1-A_{m,0})}{2A_{m,0}} \right|}{|\Re[d_1]|} = K_R.
$$

Define a set $\Omega_R$ as

$$
\Omega_R = \left\{ z \in C \,\middle|\, \Re[z] > -A - E\pi^2 \left( \frac{1}{4} + a^2 \right), \, |z| \le K_R \right\},
$$

the Brown Fixed Point Theorem implies then that $F$ has a fixed point in $\Omega_R$, i.e., there exists $\rho_m(R)$ such that

$$
\quad \rho_m(R) = F(\rho_m(R), R).
$$

At last, we prove that $\rho_m(R)$ is a continuous function of $R$ and $\Im \rho_m(R) \ne 0$. Let $G(\rho_m, R)$ be the function given by

$$
G(\rho_m, R) = F(\rho_m, R) - \rho_m.
$$

If we can prove

$$
\frac{\partial G}{\partial \rho_m} \ne 0
$$



then the Implicit Function Theorem implies that $\rho_m(R)$ is indeed a continuous function of $R$. Through (**??**), we can get that

$$
\left|\frac{\partial G}{\partial \rho_m}\right| = \left|\sum_{n=1}(-1)^{n+1}\frac{(1-A_{m,0})\,A_{m,n}}{A_{m,0}\,(1-A_{m,n})}\eta^2_{m,n}(\rho_m(R))-1\right|
$$

$$
\geq 1 - \sum_{n=1}\frac{(1-A_{m,0})\,A_{m,n}}{A_{m,0}\,(1-A_{m,n})}|\eta_{m,n}|^2
$$

$$
> 1 - \sum_{n=1}\frac{(1-A_{m,0})\,A_{m,n}\,(\Re[\rho_m(R)]+A+E\pi^2 A_{m,n})}{A_{m,0}\,(\Re[\rho_m(R)]+A+E\pi^2 A_{m,0})\,(1-A_{m,n})}|\eta_{m,n}|^2
$$

$$
= 0.
$$

To prove $\Im[\rho_m(R)] \neq 0$, we use the proof by contradiction. Direct calculation gives

$$
\frac{|\Im[d_{m,n}]|}{|\Re[d_{m,n}]|} = \frac{|2\pi\Im[\rho_m(R)]A_{m,n}-2am|}{\left|2\pi^3 E A^2_{m,n}+2\pi(A+\Re[\rho_m(R)])A_{m,n}\right|}
$$

If $\Im\rho_m(R) = 0$, we can deduce that

$$
\frac{|\Im[d_{m,n}]|}{|\Re[d_{m,n}]|} = \frac{|2am|}{\left|2\pi^3 E A^2_{m,n}+2\pi(A+\Re[\rho_m(R)])A_{m,n}\right|}
$$

$$
> \frac{|2am|}{\left|2\pi^3 E A^2_{m,n+1}+2\pi(A+\Re[\rho_m(R)])A_{m,n+1}\right|} = \frac{|\Im[d_{m,n+1}]|}{|\Re[d_{m,n+1}]|},
$$

through which and combining the continuous fraction

$$
-d_{m,0}+i = \frac{1}{d_1+\xi_{m,2}} = \frac{1}{d_{m,1}+\frac{1}{d_{m,2}+\xi_{m,3}}}
$$

we get

$$
\frac{|\Im[\eta_{m,1}]|}{|\Re[\eta_{m,1}]|} < \frac{|\Im[d_{m,1}]|}{|\Re[d_{m,1}]|},
$$

i.e.,

$$
\frac{|-\Im[d_{m,0}]+i|}{|-\Re[d_{m,0}]|} < \frac{|\Im[d_{m,1}]|}{|\Re[d_{m,1}]|} \Rightarrow
$$

$$
\frac{|-\Im[d_{m,0}]+i|}{|\Im[d_{m,1}]|} < \frac{|-\Re[d_{m,0}]|}{|\Re[d_{m,1}]|} \Rightarrow
$$

$$
< \frac{|2am+1|}{|2am|} < \frac{2\pi^3 E A^2_{m,0}+2\pi(A+\Re[\rho_m(R)])A_{m,0}}{2\pi^3 E A^2_{m,1}+2\pi(A+\Re[\rho_m(R)])A_{m,1}} < 1,
$$

which leads to a contradiction. Hence, $\Im[\rho_m(R)] \neq 0$.

*Author contributions.* All authors contribute to the writing and the analyses of this work





*Competing interests.* The authors declare that no competing interests are present

*Disclaimer.* No

*Acknowledgements.* This research was partially supported by the ONR's Young Investigator Program (N000141812588), and the Indiana University Grand Challenge Initiative. We would like to thank Dr. Shouhong Wang (Indiana University) for his various suggestions and comments on the dynamical transition framework and interpretations.

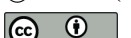



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



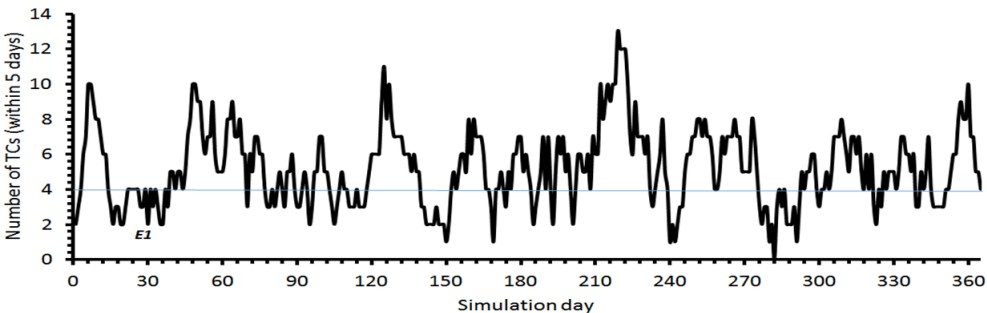

**Figure 1.** A time series of the number of TC genesis events detected each day from an idealized simulation of TC formation in a tropical channel at a homogeneous horizontal resolution of 27 km, using the Weather Research and Forecasting (WRF-ARW) model (Kieu et al. 2018).

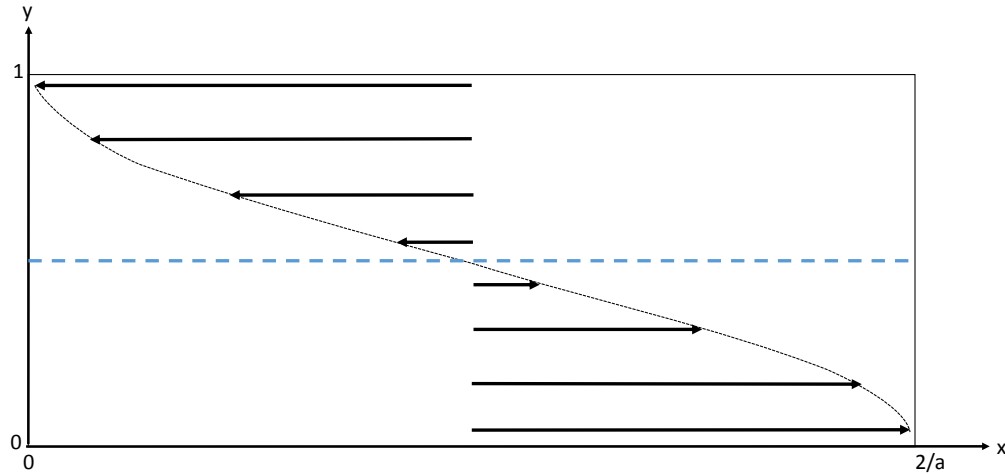

**Figure 2.** Illustration of the zonal wind that is derived from the steady-state flow $\psi_S$ in the ITCZ model (1) with the external forcing given by Eq. (3). The dotted curve represents the horizontal profile of the mean flow, while the black arrows represent the direction of the mean flow for the tropical channel domain $\Omega_a$. The blue dashed line denotes the location of the ITCZ.





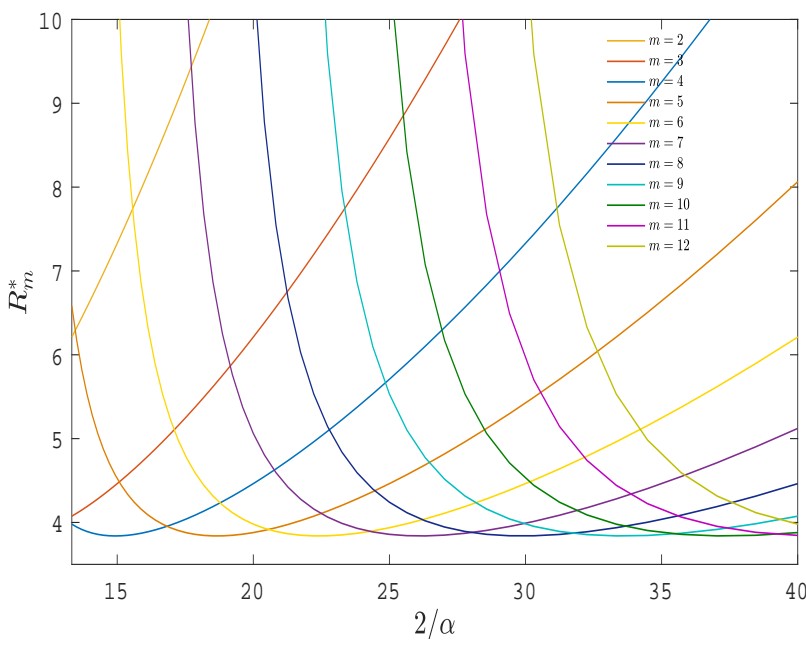

**Figure 3.** Marginal stability curves $R_m^*(a)$ obtained from the constraint on the eigenvalue $\Re\rho_{m,1}(R) = 0$ for a range of the aspect ratio $0.1 \leq a \leq 0.35$.



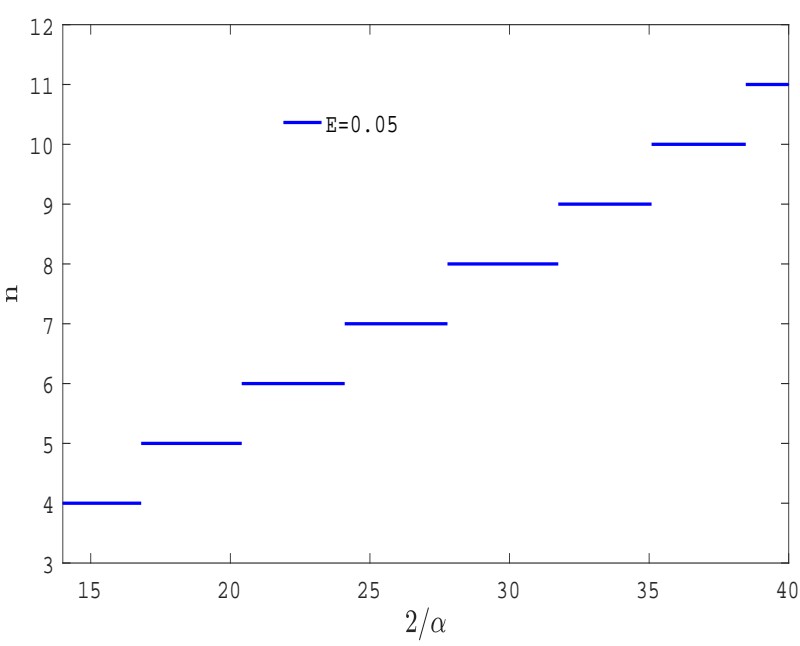

**Figure 4.** The dependence of the first critical wave number $m = n$ on the scale factor $a$, assuming the Rossby number $\epsilon = 0.5$ and the Ekman number $E = 0.05$ similar to Figure 3.





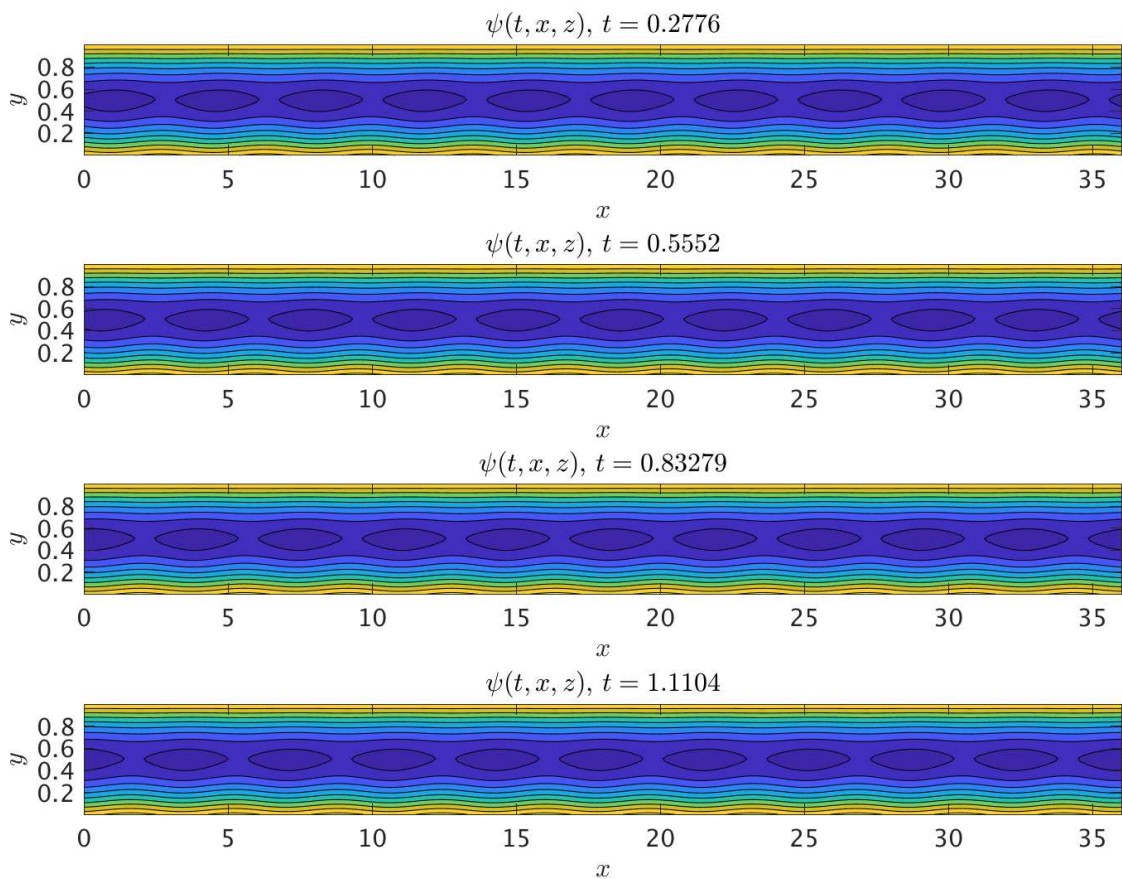

**Figure 5.** Illustration of the streamfunction $\psi$ for the new periodic state on the central manifold near the critical point $R^*$ after the dynamical transition, assuming $\epsilon = 0.3$, $E = 0.05$, and $R = 3.8717 > R^* = 3.8517$. The nondimenional period is T=2.776, which corresponds to a physical period of $\sim 3.213$ days.