# Peer review of "Large-scale Dynamics of Tropical Cyclone Formation Associated with ITCZ Breakdown"

_Atmospheric Chemistry and Physics, 2018_

## Referee Comment (RC1) · Daniel Chavas (Referee) · 9 Mar 2019

General comments: Overall this work presents a very compelling, albeit mathematically complex, physical argument in a simplified barotropic framework for the upper bound on the number of tropical cyclones that emerge from the breakdown of the ITCZ. This upper bound may have a direct role in setting the annual number of tropical cyclones on Earth, for which no current theory exists. I find the manuscript to be of very high quality in terms of both writing and physical framework, though I cannot fully evaluate the mathematical analysis, particularly for the Principle of Exchange of Stabilities, as it lies beyond the scope of my expertise. Nonetheless, if fully validated, this would appear to be a rather remarkable feat of dynamical systems theory for explaining the nature of the breakdown of the ITCZ on an Earth-like planet and its potential relevance

to global constraints on tropical cyclogenesis.

Specific comments: P2L30: It would seem very relevant here to include the work of Patricola et al. (2018; https://agupubs.onlinelibrary.wiley.com/doi/pdf/10.1002/2017GL076081) , which found that filtering out AEWs did not alter the number of storms in the Atlantic, suggesting that disturbances from other sources (e.g. ITCZ breakdown) may take their place.

P3L5/Figure 1: I'm not sure this figure, taken from Kieu et al. (2018), is appropriate here, as it is included without explanation. If the figure is presented within the stated reference then the reference alone should suffice without the figure being reproduced here.

P11L25: I'm unsure where "3 x 3000 km" is coming from here. Lx/m = 3333 km.

P11L27: I'm not sure I agree with this statement that storm size "must be larger than a limit of $\sim10^3$ km" – this may be a lower bound on size for this specific case of an equatorial band of TCs of equal size. In reality individual storms may take a range of sizes, and certainly there are instances of very small storms that appear to be have a diameter much smaller than this length scale. This is not incompatible with the model presented here, reality is simply more complex.

Given that it is an important parameter (aspect ratio, with Lx fixed), how would one plausibly define Ly for the real world? Is there some physical sense of what would represent the poleward boundary relevant to the system?

P14L34: Aren't these westward-moving disturbances simply Rossby waves? Based on Fig 2, at the ITCZ location (dashed line), Uyy=0 and thus the PV gradient is purely beta. Does their phase speed follow the barotropic Rossby wave phase speed for wavenumber equal to the unstable wavenumber predicted by the model (4 m/s)? I wonder if Rossby waves are a more appropriate analog than African Easterly Waves

for the features in the model.

To what extent does this model reproduce the behavior of the traditional model for barotropic instability under parallel shear flow and the associated Rayleigh-Kuo conditions for instability? Based on Figure 2, it appears that the physical framework is the same. However, I am trying to understand the notion that the periodic state is a secondary stable state, which is in contrast to a traditional barotropic instability model in which the instabilities would be expected to continue to grow.

Technical corrections: General: Suggest simply using "genesis" rather than "TCG" – acronyms are overused.

P4L15: might note in the text for clarity that Delta here represents the Laplacian, which is typically denoted with nabla^2

P4L17: extra "the"

P4L21: parenthesis error

P6L11: replace "while" with "though it"

P5L28: "nonlinearity"

P7L12: I believe this should be no v-wind component

P8L15: "eigenvector"

P9L13: "turns out"

P10L19: "if it exists"

P13L10: "all the same"

P14L16: "nondimensional"

P14L20: issues with the parentheses

P14L24: "obtained from the"

P14L30: "holds for the"

---

## Referee Comment (RC2) · Alex Gonzalez (Referee) · 26 Mar 2019

[referee-annotated manuscript omitted]

---

## Author Response (AR1)

**Response to Reviewer 1 (Dr. Daniel Chavas)**

We would like to thank Reviewer 1 for your constructive comments and suggestions. In this revision, we have made a number of changes to take into account your concerns. Please find below our point-by-point responses to your comments and our corresponding changes.

Overall this work presents a very compelling, albeit mathematically complex, physical argument in a simplified barotropic framework for the upper bound on the number of tropical cyclones that emerge from the breakdown of the ITCZ. This upper bound may have a direct role in setting the annual number of tropical cyclones on Earth, for which no current theory exists. I find the manuscript to be of very high quality in terms of both writing and physical framework, though I cannot fully evaluate the mathematical analysis, particularly for the Principle of Exchange of Stabilities, as it lies beyond the scope of my expertise. Nonetheless, if fully validated, this would appear to be a rather remarkable feat of dynamical systems theory for explaining the nature of the breakdown of the ITCZ on an Earth-like planet and its potential relevance.

Your encouraging evaluation of our manuscript is much appreciated. We hope that our revision below will be now acceptable to you.

**Specific comments:**
P2L30: It would seem very relevant here to include the work of Patricola et al. (2018; https://agupubs.onlinelibrary.wiley.com/doi/pdf/10.1002/2017GL076081), which found that filtering out AEWs did not alter the number of storms in the Atlantic, suggesting that disturbances from other sources (e.g. ITCZ breakdown) may take their place.

Thank you for pointing out to us this work of Patricola et al. (2018), which is indeed relevant to our study. Their finding about the insignificant role of AEWs in modulating the number of the TC climatology in the Atlantic basin is noteworthy, as it is consistent with our model presented in this study in the sense that our model does not contain any local feature such as the African Jet that triggers AEWs. One thing that we wish to note is that the AEWs are just one class of a much broader group of easterly waves in the tropical region. While AEWs can be filtered as presented in Patricola et al. (2018), the other modes of easterly waves such as mixed equatorial Robssy waves can still exist. Anyway, both Petricola et al.'s results and our study herein appear to suggest an intrinsic mechanism at the large scale, which controls the climatology of the TC numbers beyond the basin-specific features. This work has been now cited in this revision.

P3L5/Figure 1: I'm not sure this figure, taken from Kieu et al. (2018), is appropriate here, as it is included without explanation. If the figure is presented within the stated reference then the reference alone should suffice without the figure being reproduced.

Our purpose of including this figure here is to highlight the key importance of episodic development of TCs at the global scale so one can estimate the global number of TCs annually. Essentially, we need two pieces of information to be able to determine how many TCs the tropical atmosphere can support annually, which include 1) the maximum number of TCs that

the tropical atmosphere can produce for any given episode of TC formation, and 2) the frequency that the new episode of TC formation will occur. The analytical work herein addresses the first question, and the global modelling of TC formation shown in Figure 1 addresses the second question. Given that we plan to have an upcoming study that specifically tackle the second question in much more details, we have removed Figure 1 in this work per your suggestion.

P11L25: I'm unsure where "3 x 3000 km" is coming from here. Lx/m = 3333 km.

Thank you. This is our typo. We really meant 3000 km, not 3x3000 km here. This has been fixed.

P11L27: I'm not sure I agree with this statement that storm size "must be larger than a limit of ~10^3 km" – this may be a lower bound on size for this specific case of an equatorial band of TCs of equal size. In reality individual storms may take a range of sizes, and certainly there are instances of very small storms that appear to be have a diameter much smaller than this length scale. This is not incompatible with the model presented here, reality is simply more complex. Given that it is an important parameter (aspect ratio, with Lx fixed), how would one plausibly define Ly for the real world? Is there some physical sense of what would represent the poleward boundary relevant to the system?

Our main point in this discussion is that if all high zonal wavenumbers must be stable as found in this study, then any disturbance corresponding to a large wavenumber (i.e., a small size) that could potentially grow into a TC would not occur. So, only those with $m < 12$ (i.e., their diameters are $> 3000$ km) can have a chance, which agree well with the typical scale of a region where a TC emerges in the tropical region. Of course, this by no means eliminates the existence of a small TC such as midgets at the higher latitudes, because our analytical results can only provide an estimate for the size of the" hot spot" where a TC disturbance can develop. Talking about the size of a fully-developed TC is beyond our current work, as it involves various complex factors as pointed out in your several recent studies on the topic of the TC size. We have revised this discussion to avoid misleading impression.

Regarding the width of the tropical region, it is indeed hard to be precisely determined. We simply use a typical value of $20^0$ for the tropical channel, based on the definition of the tropics up to the tropic of Capricorn (~23.5°). Additional analyses for a few different widths from 15-20º do not show much difference, because the zonal scale is, after all, always an order of magnitude larger than the meridional scale.

P14L34: Aren't these westward-moving disturbances simply Rossby waves? Based on Fig 2, at the ITCZ location (dashed line), Uyy=0 and thus the PV gradient is purely beta. Does their phase speed follow the barotropic Rossby wave phase speed for wavenumber equal to the unstable wavenumber predicted by the model (4 m/s)? I wonder if Rossby waves are a more appropriate analog than African Easterly Waves for the features in the model.

We totally agree. It is entirely possible that that easterly waves here are a mode of the equatorial mixed Rossby waves, because the spatial scale as well as the phase speed are consistent with

westward-moving Rossby wave (meridional mode $n$=1). However, we also wish to caution here that the numerical procedure of finding the unstable mode on the central manifold presented in this section does not allow us to separate different modes of easterly waves. As such, the easterly waves here could be a combination of many different modes of west-ward moving Rossby waves and mixed gravity waves that we may not be able to link them specifically to the equatorial Rossby waves. This has been now mentioned in this revision.

To what extent does this model reproduce the behavior of the traditional model for barotropic instability under parallel shear flow and the associated Rayleigh-Kuo conditions for instability? Based on Figure 2, it appears that the physical framework is the same. However, I am trying to understand the notion that the periodic state is a secondary stable state, which is in contrast to a traditional barotropic instability model in which the instabilities would be expected to continue to grow.

Our discussion in the previous version was indeed unclear, which may cause some confusion here. We would like to note that the periodic state is one of the stable branches of the Hopf bifurcation if the model parameter $R$ is slightly greater than the critical value. As long as $R$ is sufficiently close to the critical number $R^*$, this periodic state on the central manifold will maintain its stable structure. For a larger $R$ number, it should be noted that the stability of the periodic state may no longer be ensured, because the central manifold function must be re-evaluated and a complex structure of the model state may arise. To some extent, this is what anticipated in the real atmosphere, because not all easterly waves can become unstable and turn into TCs. Only under some certain condition do the easterly waves become unstable.

**Technical corrections:**
General: Suggest simply using "genesis" rather than "TCG" –acronyms are overused.
Thank you. We have tried to reduce the use of the TCG acronym as suggested.

P4L15: might note in the text for clarity that Delta here represents the Laplacian, which is typically denoted with nabla^2
Thank you. The notation nabla has been now defined explicitly.

P4L17: extra "the
Corrected.

P4L21: parenthesis error
Corrected.

P6L11: replace "while" with "though it"
Modified as suggested.

P5L28: "nonlinearity"
Corrected

P7L12: I believe this should be no v-wind component
You are right. We really meant $v$-wind here. This has been now corrected.

P8L15: "eigenvector"
The typo has been corrected

P9L13: "turns out"
Modified as suggested.

P10L19: "if it exists"
This sentence has been modified

P13L10: "all the same"
Modified as suggested.

P14L16: "nondimensional"
Corrected.

P14L20: issues with the parentheses
Corrected.

P14L24: "obtained from the"
The typo has been now corrected. We thank Reviewer 1 again for your various suggestions and comments.

**Response to Reviewer 2 (Dr. Alex Gonzalez)**

We wish to thank Reviewer 2 for your encouraging comments and very detailed corrections in the annotated supplement. In this revision, we have followed your suggestions and made all necessary changes. Below please find a list of the changes that we have made in response to your comments/suggestions in the annotated PDF.

This paper revisits the well-known topic of the ITCZ breaking down due to barotropic instability into individual vortices that can be seeds for tropical cyclones. This paper provides a new and unique perspective on the topic, showing extensive mathematical derivations about the zonal wavenumber that first becomes unstable in ITCZ breakdown and how this zonal wavenumber sets constraints on the size and total number of tropical cyclones on the globe at one time. Overall, the paper is well-written and the mathematical derivations appear to be accurate. The only concerns I have are about presentation quality. There are issues with the authors being too vague about the nomenclature in their derivations, which made it difficult to check all of the math. Additionally, the authors are not consistent in physically interpreting many of the parameters, which if fixed, would help more general audiences follow the entire paper. My concerns seem like they can be addressed relatively quickly, thus I am recommending acceptance of the paper after minor revisions.

Page 1, tile: Modified as suggested.

Page 1, line 5-6: Reworded as suggested.

Page 4, line 3: Yes, this is the trade wind. The phrase has been changed to directly indicate this.

Page 4, line 15: Thank you. The definition of the Laplacian operator has been added.

Page 4. This is the total derivative, as it contains the horizontal advection component as well, i.e., $\frac{d\Delta\psi}{dt} \equiv \frac{\partial(\Delta\psi)}{\partial t} - \frac{\partial\psi}{\partial y}\frac{\partial(\Delta\psi)}{\partial x} + \frac{\partial\psi}{\partial x}\frac{\partial(\Delta\psi)}{\partial y}$ The existence of such horizontal advection is required such that the effects of background vorticity gradient can be properly taken into account as you commented (see the term that accounts for the background vorticity gradient in Eq. (8) for the contribution from the background $\widetilde{\psi_s}$.)

Page 5, table 1: Yes, the ITCZ latitude can be extended from 1500 km to 2000 km with little changes in our analyses or conclusion. This is because the meridional scale is still much less than the zonal scale, which is the circumference of the Earth (i.e., $2\pi R \sim 40,000\ km \gg 2000\ km$). Our choice of the ITCZ latitude around $12\text{-}15^0$ is simply based on the typical latitude of the ITCZ during the peak TC season.

Page 5. The unit of Horizontal eddy viscosity coefficient should be "m^2 s^-1" has been corrected.

Page 5, line 6: The phrase has been added as suggested.

Page 5, line 11, 12: changed as suggested.

Page 6, line 12: Thank you. The typos related to the domain size has been now corrected.

Page 6, line 17: we have added the definition of the non-dimensional parameter $\gamma_1$ here. You are right, this is the ratio of the vorticity forcing and vorticity response.

Page 6, line 18: reworded as suggested.

Page 6, line 23: deleted as suggested.

Page 7, line 6: You are correct. $R$ is physically a ratio between the external forcing and the sum of the viscous and linear damping terms. This physical meaning has been now included in this revision.

Page 7, line 8: the typo has been corrected.

Page 7, line 12: Thank you. This is out typo. It should be no v-wind component, not u-wind component for the boundary condition here. This has been corrected.

Page 7, line 13: delete as suggested.

Page 7, line 15: delete as suggested.

Page 7, line 16: The physical meaning of three differential operators have been now added.

Page 7, line 22: Our typo here. It should be partial derivative here after expanding all of the terms.

Page 7, line 22: this is our typo. It should be minus sign here.

Page 8, line 16: reworded as suggested.

Page 8, line 17: The way we choose the zonal wave number $m$ is very similar to the way one solves, e.g., an oscillating string held fixed between two walls. Basically, the boundary condition dictates the possible values of the eigenstates. Given the periodic boundary condition in the zonal direction, the eigenvectors include therefore all possible eigenmodes $m, \forall m \in Z^+$. So, this differs from the Fourier expansion for which there exists a pair of dual amplitudes.

Page 8, line 18: Yes, $m$ represents zonal wavenumber or meridional mode. This has now been stated clearly here.

Page 8, line 19: reworded as suggested.

Page 8, line 21: reworded as suggested.

Page 9, line 1: reworded as suggested.

Page 9, line 2: $n$ represents the order of derivative w.r.t to $y$ direction. This has been now clearly indicated.

Page 9, line 9: We have moved the definition of $\phi_{m,n}$ to the above line to make it clearer.

Page 9, line 13: reworded as suggested.

Page 9, line 17: reworded as suggested.

Page 10, equation (34): Definition of the real part operator has been now included.

Page 10, line 18: deleted as suggested.

Page 10, line 19: modified as suggested.

Page 11, line 3: corrected.

Page 11, line 5: corrected.

Page 11, line 10: reworded as suggested.

Page 11, line 14: A separated equation has been added here. Note that different value of $m$ for different value of $L_y$ is what Fig 3 is about (the value of $L_y$ is encoded in the parameter $a$), and so we don't provide a separate table for this equation.

Page 11, line 17: Per your suggestion. we have now added some comment about the result in Neito Ferreira and Schubert (1997) of the most unstable zonal wavenumber being wavenumber 13 here.

Page 11, line 22: changed as suggested.

Page 11, line 23: reworded  as suggested.

Page 11, line 25, 26: This is out typo. It should be 3000 km, not 3x3000 km. This has been fixed.

Page 11, line 27: Is there a citation to back up this statement? "that has been long observed but not fully explained so far."

Page 12, line 3: the Rayleigh and/or Fjørtoft necessary conditions for instability has been now explicitly mentioned in this revision.

Page 12, line 3: reworded as suggested.

Page 13, line 10: reworded as suggested.

Page 14, line 7: reworded as suggested.

Page 14, line 13: Thank you for commenting on this. Because $\nu\pi^4 \ll \alpha L^2\pi^2$ in our calculation of $R$ (see Eq 9), use of $\nu = 100, or\ 1000$ m$^2$ s$^{-1}$ would not change much our estimation of the critical number . We have now mentioned this explicitly in this work.

Page 14, line 28: turn "this new state" to "the disturbances in this new state".

Page 14, line 31: Thank you. The follow vectors have been added in this revision.

Page 15, line 8: changed to "feedbacks".

Page 15, line 11: changed to "discussed hereinafter".

Page 15, line 14: delete "," after (TCs).

Page 15, line 15: corrected as suggested.

Page 15, line 17: reworded as suggested.

Page 15, line 19: reworded as suggested.

Page 15, line 21: reference might help here "Using the Principle of Exchange of Stabilities condition for the ITCZ model"

Page 15, line 24: typo, add "." after generate.

Page 15, line 25: "k~12" = (the word estimation already implies an approximation).

Page 16, line 10: reworded as suggested.

Page 17, line 5: fixed.

Page 19, line 1: fixed.

Figure 3, 4: $m = 13$ has been added. The label axis has been also fixed.

Figure 5: Fixed.

```
%% Copernicus Publications Manuscript Preparation Template for LaTeX
Submissions
%% ---------------------------------
%% This template should be used for copernicus.cls
%% The class file and some style files are bundled in the Copernicus
Latex Package, which can be downloaded from the different journal
webpages.
%% For further assistance please contact Copernicus Publications at:
production@copernicus.org
%%
https://publications.copernicus.org/for_authors/manuscript_preparation.ht
ml
%% Please use the following documentclass and journal abbreviations for
discussion papers and final revised papers.
%% 2-column papers and discussion papers
\documentclass[journal abbreviation, manuscript]{copernicus}

\usepackage{amssymb, amsmath, amsfonts}
\usepackage{epstopdf}
\usepackage{booktabs}

%% Journal abbreviations (please use the same for discussion papers and
final revised papers)
% Advances in Geosciences (adgeo)
% Advances in Radio Science (ars)
% Advances in Science and Research (asr)
% Advances in Statistical Climatology, Meteorology and Oceanography
(ascmo)
% Annales Geophysicae (angeo)
% Archives Animal Breeding (aab)
% ASTRA Proceedings (ap)
% Atmospheric Chemistry and Physics (acp)
% Atmospheric Measurement Techniques (amt)
% Biogeosciences (bg)
% Climate of the Past (cp)
% DEUQUA Special Publications (deuquasp)
% Drinking Water Engineering and Science (dwes)
% Earth Surface Dynamics (esurf)
% Earth System Dynamics (esd)
% Earth System Science Data (essd)
% E&G Quaternary Science Journal (egqsj)
% Fossil Record (fr)
% Geographica Helvetica (gh)
% Geoscientific Instrumentation, Methods and Data Systems (gi)
% Geoscientific Model Development (gmd)
% History of Geo- and Space Sciences (hgss)
% Hydrology and Earth System Sciences (hess)
% Journal of Micropalaeontology (jm)
% Journal of Sensors and Sensor Systems (jsss)
% Mechanical Sciences (ms)
% Natural Hazards and Earth System Sciences (nhess)
% Nonlinear Processes in Geophysics (npg)
```

```
% Ocean Science (os)
% Primate Biology (pb)
% Proceedings of the International Association of Hydrological Sciences
(piahs)
% Scientific Drilling (sd)
% SOIL (soil)
% Solid Earth (se)
% The Cryosphere (tc)
% Web Ecology (we)
% Wind Energy Science (wes)

%% \usepackage commands included in the copernicus.cls:
%\usepackage[german, english]{babel}
%\usepackage{tabularx}
%\usepackage{cancel}
%\usepackage{multirow}
%\usepackage{supertabular}
%\usepackage{algorithmic}
%\usepackage{algorithm}
%\usepackage{amsthm}
%\usepackage{float}
%\usepackage{subfig}
%\usepackage{rotating}

\begin{document}
\title{Large-scale Dynamics of Tropical Cyclone Formation Associated with
ITCZ Breakdown}

% \Author[affil]{given_name}{surname}

\Author[1]{Quan}{Wang}
\Author[2,*]{Chanh}{Kieu}
\Author[2]{The-Anh}{Vu}
\affil[1]{Department of Mathematics, Sichuan University, Sichuan, China}
\affil[2]{Department of Earth and Atmospheric Sciences, Indiana
University, Bloomington, IN 47405}
%
% The [] brackets identify the author with the corresponding affiliation.
1, 2, 3, etc. should be inserted.
%
\runningtitle{Dynamics of Tropical Cyclone Formation}
\runningauthor{Kieu et al.}
\correspondence{ckieu@indiana.edu}
\received{}
\pubdiscuss{} %% only important for two-stage journals
\revised{}
\accepted{}
\published{}
%
% These dates will be inserted by Copernicus Publications during the
typesetting process.
%
\firstpage{1}
```

Mathematics, Sichuan University,
Sichuan, China}¶

[revised manuscript text omitted]

Although the ITCZ breakdown appears to be a slow process as compared to other pathways such as vortex merger \citep[e.g.,][]{WangMagnusdottir2006, KieuZhang2008, KieuZhang2010} or tropical easterly waves \citep[e.g.,][]{Zehnder_etal1999, Molinari_etal1997, Halverson_etal2007, Dunkerton_etal2009, Montgomery_etal2010, Wangetal2012}, it is an inherent property of the tropical atmosphere at the global scale that could provide a source of large-scale disturbances responsible for TCG. To minimize the complication due to the basin-specific features, we thus limit our study

of the global TC formation to an idealized aqua-planet configuration to facilitate the analytical analyses in this study.

The rest of the paper is organized as follows. In the next section, an analytical model for the large-scale TC genesis based on the ITCZ breakdown model is presented. Section 3 presents detailed analyses of the principle of exchange of stabilities for the ITCZ model as well as the stability analyses of the dynamical transition. Numerical examination will be discussed in Section 4, and concluding remarks are given in the final section.
%
% Section
%
\section{Formulation}
A unique characteristic of the ITCZ that provides a favorable environment for genesis to occur is the highly unstable zone along the ITCZ where trade winds from the two hemispheres converge. Such a zone with strong horizontal shear is well documented along the tropical belt where the potential vorticity gradient changes sign, providing a necessary condition for disturbances to grow according to Rayleigh's theorem \citep{CharneyStern1962, FerreiraSchubert1997}. Thus, a disturbance embedded within in the ITCZ can trigger a nonlinear growth and extract the energy from the background, resulting in a potential amplification of the disturbance.

Because of such a dominant role of the ITCZ in the global TC formation, a natural model for TCG should take into account the large-scale ITCZ breakdown processes. This ITCZ breakdown model is particularly suitable for an aqua-planet that does not have other triggering mechanisms such as land-sea interaction or terrain effects. For this reason, we will consider the ITCZ breakdown as a starting model for the TC genesis at the global scale in this study. Inspired by the modeling studies of the ITCZ breakdown based on the shallow water equation by \cite{FerreiraSchubert1997}, we examine a similar model for the ITCZ dynamics on a horizontal plane in which the governing equation for the ITCZ can be reduced to an equation for the potential vorticity as follows

\begin{align}\label{eq1}
\frac{d \Delta \psi}{dt}=\nu_{e}\Delta^{2}\psi +F-\alpha\Delta\psi-\beta \frac{\partial\psi}{\partial x},
\end{align}
where the horizontal streamfunction $\psi$ has been introduced as a result of the continuity equation, $\nu_{e}$ is horizontal eddy viscosity, $\alpha$ is a relaxation time, $\Delta$ is the Laplacian operattor, and $F$ is an external force that represents the either a source/sink of mass within the ITCZ or vorticity source \footnote{In \cite{CharneyDeVore1979}, the relaxation time $\alpha$ is proportional to the ratio of the Ekman depth $D_E$ over the depth of the fluid $H$, while the external forcing term $F$ can be treated as a large-scale vorticity source.}. Note here that the derivative on the left hand side of Eq. \eqref{eq1} is the total derivative such that the horizontal advection of the vorticity is included. As discussed in \cite{FerreiraSchubert1997}, the mass source/sink term $F$ is important for the ITCZ dynamics, because the horizontal dynamics could not fully capture the vertical mass flux

within the ITCZ. Unlike the original ITCZ model in
\cite{FerreiraSchubert1997}, we have however introduced in the above
model \eqref{eq1} an explicit drag forcing term to represent the impacts
of eddy diffusion as discussed in \cite[e.g.,][]{RambaldiMo1984,
LegrasGhil1985, FerreiraSchubert1997}. The governing equation \eqref{eq1}
for the horizontal streamfunction has been extensively used in previous
studies to examine the quasi-geostrophic dynamics under different large-
scale conditions \citep[e.g.,][]{CharneyDeVore1979, LegrasGhil1985,
RambaldiMo1984, Schar1990}.

To be specific for our TCG problem, we will apply Eq. \eqref{eq1} for a
zonally periodic tropical channel, which is defined as
\begin{align}\label{eq2}
\Omega=\left[0,L_{x}\right]
\times\left[0,L_{y}\right],
\end{align}
where $L_{y}$ is the width of the tropical channel in a hemisphere and
$L_{x}$ is the zonal length of the channel. This domain roughly
represents a region where the ITCZ can be treated as a long band wrapping
around the Equator. For the current Earth condition, $L_x \sim
40,000$ km, and $L_y \sim 1,000-1,500$ km (i.e., 10-15$^o$), and so by
definition $L_x \gg L_y$.
%
% Table 1
%
\begin{table}[t]
\caption{Parameters of the model}\label{t1}
\begin{tabular}{cll}
\tophline
$Variable$ & $Range$ & $Remark$\\
\middlehline
$U_o$ & 10-20 m s$^{-1}$ & Mean easterly flow in the tropical lower
troposphere \\
$L_y$ & 1200-1500 km & Width of the tropical channel $\Omega$ \\
$L_x$ & $\sim$40,000 km & Length of the tropical channel domain
$\Omega$\\
$a$ & $\frac{2L_y}{L_x}$ & Aspect ratio of the tropical channel \\
$\alpha$ & $ 10^{-5}-10^{-7} \mbox{s}^{-1}$ & Relaxation time \\
$\nu$ & $10-10^{4} \mbox{m}^{2} \mbox{s}^{-1}$ & Horizontal eddy
viscosity coefficient \\
$\beta$ & $2\times 10^{-11} \mbox{s}^{-1}$ & Variation of the Coriolis
parameter with latitudes \\
$\gamma$ & $10^{-10}-10^{-11} \mbox{s}^{-2}$ & Magnitude of the external
mass source/sink in the ITCZ breakdown model \\
\bottomhline
\end{tabular}
\end{table}

Before we can analyze the ITCZ breakdown model, it is necessary to have
first an explicit expression for the forcing term $F$. In the early study
by \cite{FerreiraSchubert1997}, $F$ represents a mass sink that is a
piecewise unit step function of latitudes. To account for the existence
of the zonal jet in mid-latitude regions, \cite{LegrasGhil1985} however
used a forcing of the form $F=\alpha \nabla \psi^*$, where $\psi^*$ is a

given streamfunction that represents the zonal jet around $50^o N$. Given our focus on the ITCZ dynamics, we will choose this forcing term such that its corresponding steady state can best represent the typical background flow in the tropical lower troposphere. A zonally symmetric functional form for the $F$ that meets this requirement is
\begin{align}\label{eq2}
F=\gamma\sin\frac{\pi y}{L_{y}}
\end{align}
where $\gamma$ denotes the strength of the forcing. Note that this forcing amplitude is not arbitrary, because its value dictates the zonal mean flow in the tropical region as will be shown below.

While the forcing term given by Eq. \eqref{eq2} differs from the unit step function in \cite{FerreiraSchubert1997}, it turns out that \eqref{eq2} allows a steady solution consistent with the typical flow near the ITCZ. Indeed, the steady state solution $\psi_S$ of \eqref{eq1} that results from this zonally symmetric forcing is
\begin{align}\label{steady-state0}
\psi_{S}=\frac{-\gamma L_{y}^{4}}{\nu_{e}\pi^4+\alpha
L_{y}^{2}\pi^2}\sin\frac{\pi y}{L_{y}}.
\end{align}
The horizontal flow corresponding to this steady streamfunction is illustrated in Fig. \ref{fig2}, which shows two opposite easterly and westerly flows to the north and the south of an ITCZ during a typical Northern Hemisphere summer as expected.

Given the above forcing $F$ and its corresponding steady state, the problem of the ITCZ breakdown is now mathematically reduced to the study of the instability of the steady-state \eqref{steady-state0} as the model parameters such as the forcing amplitude $\gamma$, the relaxation time $\alpha$, or the beta effect vary. To this end, it is more convenient to re-write Eq. \eqref{eq1} in the non-dimensional form such that our subsequent mathematical analyses can be simplified. Given the governing equation \eqref{eq1}, it is apparent that the natural scaling for time, streamfunction, and distance can be chosen respectively as follows:
\begin{align*}
t=\frac{1}{L_y \beta}t^*,~~
\psi=L_y U_{0}\psi^*,
F=\frac{\alpha U_{0}}{L_{y}}F^*,
\end{align*}
where the asterisk denotes the nondimensional variables, and $U_{0}$ is a given characteristic horizontal velocity that determines the strength of the zonal mean flow in the tropical region. Nondimensionalizing Eq. \eqref{eq1} and neglecting the asterisks hereinafter, the nondimensional form for Eq. \eqref{eq1} becomes
\begin{align}\label{eq3}
\frac{\partial\Delta \psi}{\partial t}+\epsilon
J(\psi,\Delta\psi)=&E\Delta^{2}\psi+
F-A\Delta\psi
-\frac{\partial\psi}{\partial x},
\end{align}
where
\begin{align*}
&\epsilon=\frac{U_{0}}{L_y^2\beta}~~ \text{is the Rossby number,} \\

```
&E=\frac{\nu_{e}}{L_y^{3}\beta}~~ \text{is the Ekman number,}\\
&A=\frac{\alpha}{L_y\beta} ~~\text{is the ratio of the relaxation time to
the inherent time related to the Earth's rotation rate}.
\end{align*}
```

For the sake of mathematical convenience, we will hereinafter extend the
model domain from $[0,L_y]$ to $[-L_y,L_y]$ such that the boundary
conditions become meridionally symmetric along the Equator at $y=0$. This
mathematical method of extending the domain will simplify a lot of
calculations, though it has no effect on our solutions so long as we
limit the final solution in the original domain $[0,L_x]\times
[0,L_y]$ and maintain the Neumann boundary at $y=0$ as shown below. The
non-dimensional extended domain is therefore given by

```
\begin{align*}
\Omega=\left[0,\frac{2}{a}\right]
\times\left[-1,1\right].
\end{align*}
```
where the scale factor $a\equiv 2L_y/L_x$ is introduced to simplify our
spectral analyses. Given the above nondimensionlization, the non-
dimensional form of the forcing \eqref{eq2} is now simply
```
\begin{align}\label{eq4}
F=\gamma_{1}\sin\pi y,
\end{align}
```
where the nondimensional parameter $\quad\gamma_{1}=\frac{\gamma
L}{\alpha U_{0}}$ denotes the ratio of the vorticity forcing amplitude
$\gamma$ to the vorticity response $U_0$, and the non-dimensional form of
the steady state \eqref{steady-state0} is

\quad\gamma_{1}=\frac{\gamma L}{\alpha U_{0}},

```
\begin{align}\label{steady-state}

[revised manuscript text omitted]

\begin{align*}
\phi_{0,n}=0,~~\widetilde{\phi}_{0,n}=0,\qquad n\geq0,
\end{align*}
and so there would exist no solution at all, which contradicts our
assumption of the existence of the eigenvector for $m=0$. Thus, the
zonally symmetric mode with $m=0$ is always stable. Because this stable
mode does not allow us to examine any transition behaviors, this special
mode will not be considered hereafter.

For $m\neq 0$, it can be seen also from \eqref{real-part} that all
possible unstable eigenvectors with $m\ne0$ must satisfy the following
constraints
\begin{align}\label{eq:constraint}
\begin{cases}
\begin{cases}
\phi_{m,n}=0,~~\text{when}, ~ a\geq\frac{\sqrt{3}}{2},~m\in Z;\\
\phi_{m,n}=0,~~\text{when},~ \frac{\sqrt{3}}{4}\leq
a<\frac{\sqrt{3}}{2},~|m|\geq2,\\
\phi_{m,n}=0,~~\text{when},~ \frac{\sqrt{3}}{6}\leq
a<\frac{\sqrt{3}}{4},~|m|\geq3,\\
\cdots\qquad\vdots\qquad\cdots\\
\phi_{m,n}=0,~~\text{when},~ \frac{\sqrt{3}}{2k}\leq
a<\frac{\sqrt{3}}{2k-2},~|m|\geq k
\end{cases}\\
\widetilde{\phi}_{m,n}=0,\text{for all} \quad a>0.
\end{cases}
\end{align}
This conclusion can be explicitly confirmed if we note again that the
constraint \eqref{eq:constraint} will ensure that the coefficient
$A_{m,n}>0$, and $B_{m,n}<0$. If we assume that there exists any unstable
eigenvector $\psi_m$ with some wavenumber $m \ne 0$ such that the
corresponding eigenvalue $\rho_m$ satisfies $\Re[\rho_m]>0$, then Eq.
\eqref{real-part} immediately indicates that $\phi_{m,n}=0, \forall~ |m|
\ge k$ and $n\in\mathbb{Z}^{+}\cup\{0\}$ (i.e., $\psi_m=0$), and so no
such unstable eigenvector $\psi_m$ can exist at all. As a result, we
obtain a remarkable result that any possible unstable modes must be
constrained by the condition $|m|\le k$, where $k$ is an integer
satisfying the following relationship
\begin{equation}\label{maxm}
\frac{\sqrt{3}}{2k}\leq a<\frac{\sqrt{3}}{2k-2}.
\end{equation}

To help understand the significance of this result, we consider a
tropical channel domain between 10$^o$S-10$^o$N in the Earth's atmosphere
(i.e., $L_y \sim 1200$ km) and $L_x \sim 40,000$ km such that $a \equiv
2L_y/L_x \approx 0.06$. Using the condition $\frac{\sqrt{3}}{2k}\leq
a<\frac{\sqrt{3}}{2k-2}$, we obtain an upper bound wavenumber $k \approx

$12$. As can be seen from \eqref{maxm}, a narrower tropical channel width (i.e., smaller $L_y$) would lead to a smaller the scale ratio $a$, and so the upper bound $k$ will be higher. For our typical tropical region, the largest integer number $m \le 12$ is consistent with the most unstable zonal mode $m=13$ obtained from the modelling study in \cite{FerreiraSchubert1997}. %One could in principle choose the tropical width $L_y$ such that the upper bound $k$ perfectly matches with the most unstable zonal wavenumber 13 as reported in \cite{FerreiraSchubert1997}. Although we do not know exactly in advance which wavenumber $m < k$ will become unstable, because the condition $|m| < k$ includes a range of $m$ whose real part $\Re[\rho_m]$ could be positive, the above result is still very significant due to its explicit indication that the unstable wavenumbers cannot be arbitrary but must be bounded. Any eigenvectors with $|m| \geq k$ must be therefore stable and cannot grow.

[revised manuscript text omitted]

%Using numerical analyses, it was also found that as the parameter $R$ increases, there are one pair (or two pairs) of complex conjugate eigenvectors that becomes first critical. In the case with one pair of complex conjugate eigenvectors becoming first critical, the transition is either continuous or catastrophic as described by a Hopf bifurcation depending on the sign of a single non-dimensional transition number. For

this case, our numerical results  showed that the system exhibits only super-critical Hopf bifurcations, i.e. there exists a stable time-periodic solution  associated with the mean flow \eqref{steady-state0}. For the case of two pairs of complex conjugate eigenvectors becoming first critical, there are many possible and interesting transition scenarios depending on the relations between four nondimensional transition parameters, which in turn depend on the nonlinear interactions of the critical modes with the stable modes. This case is non-generic and occurs when the control parameter crosses a co-dimension two critical point, which is only possible in multi parameter systems such as the one considered here. Such a critical point lies at the transversal intersection of two neutral stability curves of Hopf bifurcations and is known as double Hopf (or Hopf-Hopf) bifurcation in the bifurcation theory literature. In this case, our numerical experiments suggest that the system admits only one possible transition scenario and it is continuous type transitions as in the (single) Hopf case. Moreover, we prove that an $S^3$ -attractor bifurcates when this co-dimension two critical point is crossed in the parameter space. Under the restricted parameter regime dictated by our numerical observations, it is shown that the bifurcated attractor has a limit cycle or an invariant torus as a repellor.

%% The following commands are for the statements about the availability of data sets and/or software code corresponding to the manuscript.
%% It is strongly recommended to make use of these sections in case data sets and/or software code have been part of your research the article is based on.
%\codeavailability{TEXT} %% use this section when having only software code available
%\dataavailability{TEXT} %% use this section when having only data sets available
%\codedataavailability{TEXT} %% use this section when having data sets and software code available
%\sampleavailability{TEXT} %% use this section when having geoscientific samples available

\appendix
\section{Principle of Exchange of Stabilities}
The Principle of Exchange of Stabilities (PES) for a dynamical system basically refers to a critical condition for which the eigenvalues of the linear operator first cross a prescribed value. More precisely, the PES can be precisely stated as follows.

Let $\mathbf{L}_{\lambda}$ and $\mathbf{G}$ represent the linear and nonlinear parts of a dynamical system in the abstract form:
\begin{equation}\label{eq_a1}
\frac{d\mathbf{u}}{dt} = \mathbf{L}_{\lambda}(\mathbf{u}) + \mathbf{G}(\mathbf{u},\lambda)
\end{equation}
where ${\lambda} \in \mathbb{R}$ is the model parameter, and $\mathbf{u} \in \mathbb{R}^n$ represents the state of the system. By defniition, $\mathbf{L}_\lambda$ is a parameterized linear operator that depends continuously on $\lambda$. Consider the eigenvalue equation given by
\begin{equation}\label{eq_a2}
\mathbf{L}_{\lambda} \mathbf{e} = \beta(\lambda)\mathbf{e},

```latex
\end{equation}
where $\mathbf{e}$ is eigenvector, and $\beta(\lambda) \in
\mathbb{C}$ the eigenvalue. Let ${\beta_j (\lambda) \in \mathbb{C} | j
\in \mathbb{N}}$ be the eigenvalues (counting multiplicity) of
$\mathbf{L}_\lambda$. If we have
\begin{equation} \label{eq_a3}
   \begin{cases}
  \Re[\beta_j (\lambda)]
  \begin{cases}
  <0 & \text{ if } \lambda <\lambda_0,\\
  =0 & \text{ if } \lambda=\lambda_0, \; \forall 1\le i \le m\\
  >0 & \text{ if } \lambda<\lambda_0,
  \end{cases},\\
  \Re[\beta_j(\lambda_0)] <0, \; \forall j \ge m+1,
\end{cases}
\end{equation}
then the system is said to satisfy the PES condition at $\lambda_0$,
which signifies a bifurcation of the system from one state to another.
For dissipative systems, the PES condition has a much more powerful
implication than a simple bifurcation, as it ensures a dynamical
transition that can be completely categorized by three different types of
transition including the continuous transition, the catastrophic
transition, and the random transition. See \cite{MaWang2013} for more
details of the PES conditions for nonlinear systems.
%
% Appendix 2
%
\section{Existence of the critical number $R^*$}
For $\frac{\sqrt{3}}{2k+2}\leq a<\frac{\sqrt{3}}{2k}$ and $1\leq m\leq
k~(k=1,2,\cdots)$, it is easy to see from \eqref{real-part} that we must
have
\begin{align*}
\phi_{m,0}\neq0,
\end{align*}
because otherwise we will have $\phi_{m,n}=0,n\geq0$, and there would
exist no eigenvectors. For the sake of convenience, we will hereinafter
replace $\psi_{m}$ by $\frac{\psi_{m}}{B_{m,0}\phi_{m,0}}$, and similarly
replace $\phi_{m,n}$ by $\frac{\phi_{m,n}}{B_{m,0}\phi_{m,0}}$.
%Let's denote
%\begin{align*}
%\phi_{m,n}'=\frac{\phi_{m,n}}{B_{m,0}\phi_{m,0}},~n\geq0,
%\end{align*}
%and omitting primes,
Equations \eqref{eq17a}-\eqref{eq17b} are then re-written as follows:
\begin{align}\label{relation0}
\begin{aligned}
&B_{m,n+1}\phi_{m,n+1}+C_{m,n}\phi_{m,n}
-B_{m,n-1}\phi_{m,n-1}=0,~~n\geq1,\\
&B_{m,1}\phi_{m,1}+C_{m,0}\phi_{m,0}
+i\left(A_{m,0}\phi_{m,0}-
\phi_{m,0}\right)=0, ~~n=0,
\end{aligned}
\end{align}
and
```

```
\begin{align}\label{real-part2}
&\sum_{n\geq0}B_{m,n}
A_{m,n}(E\pi^2A_{m,n}+A+\Re[\rho_m])
|\phi_{m,n}|^{2}=0.
\end{align}
Denote
\begin{align*}
d_{m,n}=\frac{C_{m,n}}{B_{m,n}},
\end{align*}
and let
\begin{align*}
\eta_{m,n}=B_{m,n}\phi_{m,n},
\end{align*}
\eqref{relation0} can be further rewritten as
\begin{align}\label{relation1}
\begin{aligned}
&\eta_{m,n+1}+d_{m,n}\eta_{m,n}
-\eta_{m,n-1}=0,\quad n\geq1,\\
&\eta_{m,1}+d_{m,0}
-i=0,\quad n\geq0.
\end{aligned}
\end{align}
This reduced equation \eqref{relation1} allows us to deduce a number of
important constraints. Indeed, we re-arrange \eqref{relation1} as
follows:
\begin{align}\label{relation2}
\begin{aligned}
&-d_{m,0}+i=\eta_{m,1},\quad\eta_{m,0}=1,\\
&\xi_{m,n}=\frac{\eta_{m,n}}{\eta_{m,n-1}}=\frac{1}
{d_{m,n}+\frac{\eta_{m,n+1}}{\eta_{m,n}}},\\
&-d_{m,0}+i=\frac{1}{d_{m,1}+\xi_{m,2}}=
\frac{1}{d_{m,1}+\frac{1}{d_{m,2}+\xi_{m,3}}}.
\end{aligned}
\end{align}
It is readily seen from \eqref{relation1} that $\eta_{m,n}=0$ for all
$n\geq0$ whenever there exists a $l \geq 0$ for which $\eta_{m,l}=0$.
This means that $\xi_{m,n}\neq0$ for all $n \geq 0$. From
\eqref{relation2} one can derive that
\begin{align}\label{expression}
\eta_{m,n}\equiv \xi_{m,1}
 \xi_{m,2}\cdots\xi_{m,n},~n\geq1.
\end{align}
Therefore, for $
\frac{\sqrt{3}}{2k+1}\leq a<\frac{\sqrt{3}}{2k}~(k=1,3,3,...)$,
\eqref{real-part2} can be equally rewritten as:
\begin{align}\label{equality}
\begin{cases}
\sum_{n\geq0}\Re [d_{m,n}]|\eta_{m,n}|^{2}=0,\\
\Re [d_{m,n}]<0(n\geq1), \Re[d_{m,0}]>0,
\end{cases}\quad
 m\leq k.
\end{align}
One can deduce from the third equality of \eqref{relation2} that
\begin{align}\label{eigenvalue11}
```

```
\rho_{m}=-A-\pi^{2}EA_{m,0}
+\frac{2iam+iam\pi^2 R\left(1-A_{m,0}\right)}{2\pi A_{m,0}}+\frac{\frac{-
am\pi R\left(1-A_{m,0}\right)}{2A_{m,0}}}{d_{1}+\xi_{m,2}}.
\end{align}
Let's define a function $F$ using right hand side of
\eqref{eigenvalue11}, i.e.,
\begin{align*}
F(\rho_{m},R)=-A-\pi^{2}EA_{m,0}
+\frac{2iam+iam\pi^2 R\left(1-A_{m,0}\right)}{2\pi A_{m,0}}+\frac{\frac{-
am\pi R\left(1-A_{m,0}\right)}{2A_{m,0}}}{d_{1}+\xi_{m,2}}.
\end{align*}
Due to the fact that
\begin{align*}
&\Re d_{m,n}<0(n\geq1),
\end{align*}
we can obtain that
\begin{align*}
|F(\rho_{m},R)|\leq&\left|-A-\pi^{2}EA_{m,0}
+\frac{2iam+iam\pi^3 R\left(1-A_{m,0}\right)}{2\pi
A_{m,0}}\right|\\&+\frac{\left|\frac{-am\pi
 R\left(1-A_{m,0}\right)}{2A_{m,0}}\right|}{|\Re [d_{1}]|}=K_{R}.
\end{align*}

\noindent Define a set $\Omega_{R}$ as
\begin{align*}
\Omega_R=\left\{z\in C\Bigg| \Re [z]>-A-
E\pi^2\left(\frac{1}{4}+a^2\right),~|z|\leq K_R\right\},
\end{align*}
the Brown Fixed Point Theorem implies then that $F$ has a fixed point in
$\Omega_{R}$, i.e., there exists $\rho_{m}(R)$
such that
\begin{align*}
\rho_{m}(R)=F(\rho_{m}(R),R).
\end{align*}

At last, we prove that $\rho_{m}(R)$ is a continuous function of $R$ and
$\Im\rho_{m}(R)\neq0$.
Let $G(\rho_{m},R)$ be the function given by
\begin{align*}
G(\rho_{m},R)=F(\rho_{m},R)-\rho_{m}.
\end{align*}
If we can prove
\begin{align*}
\frac{\partial G}{\partial \rho_{m}}\neq0
\end{align*}
then the Implicit Function Theorem implies that $\rho_{m}(R)$ is indeed a
continuous function of $R$. From the definition of $G$ and
\eqref{eigenvalue11}, we obtain
\begin{align*}
\left|\frac{\partial G}{\partial \rho_{m}}\right|
&=\left|\sum_{n=1}(-1)^{n+1}\frac{\left(1-A_{m,0}\right)A_{m,n}}{A_{m,0}
\left(1-A_{m,n}\right)}\eta_{m,n}^{2}(\rho_{m}(R))
-1\right|\\
&\geq1-\sum_{n=1}\frac{\left(1-A_{m,0}\right)A_{m,n}}
```

continuous function of $R$.¶
Through \eqref{est1}, we can get that¶

```latex
{A_{m,0} \left(1-A_{m,n}\right)}|\eta_{m,n}|^{2}\\
&>1-\sum_{n=1}\frac{\left(1-A_{m,0}\right)A_{m,n}
\left(\Re[\rho_{m}(R)]+A+E\pi^2A_{m,n}\right)}{A_{m,0}
\left(\Re[\rho_{m}(R)]+A+E\pi^2A_{m,0}\right) \left(1-
A_{m,n}\right)}|\eta_{m,n}|^{2}\\
&=0.
\end{align*}
To prove $\Im[\rho_{m}(R)]\neq0$, we use the proof by contradiction.
Direct calculation gives
\begin{align*}
\frac{\left|\Im [d_{m,n}]\right|}{\left|\Re[ d_{m,n}]\right|}
=\frac{\left|2\pi \Im[\rho_{m}(R)] A_{m,n}-
2am\right|}{\left|2\pi^{3}EA^{2}_{m,n}
+2\pi(A+\Re[\rho_{m}(R)])A_{m,n}\right|}
\end{align*}
If $\Im\rho_{m}(R)=0$, we can deduce that
\begin{align*}
&\frac{\left|\Im [d_{m,n}]\right|}{\left|\Re [d_{m,n}]\right|}
=\frac{\left|2am\right|}{\left|2\pi^{3}EA^{2}_{m,n}
+2\pi(A+\Re[\rho_{m}(R)])A_{m,n}\right|}\\&>\frac{\left|2am\right|
}{\left|2\pi^{3}EA^{2}_{m,n+1}
+2\pi(A+\Re[\rho_{m}(R)])A_{m,n+1}\right|}=\frac{\left|\Im
[d_{m,n+1}]\right|}{\left|\Re [d_{m,n+1}]\right|},
\end{align*}
through which and combining the continuous fraction
\begin{align*}
-d_{m,0}+i=\frac{1}{d_{1}+\xi_{m,2}}=
\frac{1}{d_{m,1}+\frac{1}{d_{m,2}+\xi_{m,3}}}
\end{align*}
we get
\begin{align*}
\frac{\left|\Im [\eta_{m,1}]\right|}{\left|\Re
[\eta_{m,1}]\right|}<\frac{\left|\Im [d_{m,1}]\right|}{\left|\Re
[d_{m,1}]\right|},
\end{align*}
i.e.,
\begin{align*}
&\frac{\left|-\Im [d_{m,0}]+i\right|}{\left|-\Re
[d_{m,0}]\right|}<\frac{\left|\Im [d_{m,1}]\right|}{\left|\Re
[d_{m,1}]\right|}\Rightarrow\\
&\frac{\left|-\Im [d_{m,0}]+i\right|}{\left|\Im
[d_{m,1}]\right|}<\frac{\left|-\Re [d_{m,0}]\right|}{\left|\Re
[d_{m,1}]\right|}\Rightarrow\\
&1<\frac{\left|2am+1\right|}{\left|2am\right|}
<\frac{2\pi^{3}EA^{2}_{m,0}
+2\pi(A+\Re[\rho_{m}(R)])A_{m,0}}{2\pi^{3}EA^{2}_{m,1}
+2\pi(A+\Re[\rho_{m}(R)])A_{m,1}}<1,
\end{align*}
which leads to a contradiction. Hence, $\Im[\rho_{m}(R)]\neq0.$

\noappendix        %% use this to mark the end of the appendix section
\authorcontribution{CK perceived the model for TC genesis. QW carried out
mathematical proofs related to transition dynamics. AV conducted
```

[revised manuscript text omitted]

\bibitem[{Montgomery et~al.(2010)Montgomery, Wang,, and
  Dunkerton}]{Montgomery_etal2010}

Montgomery, M.~T., Wang, Z., and Dunkerton, T.~J.: Coarse, intermediate and
  high resolution numerical simulations of the transition of a tropical wave
  critical layer to a tropical storm. Atmospheric Chemistry and
  Physics, 10, 10\,803--10\,827,
  https://doi.org/10.5194/acp-10-10803-2010,
  \urlprefix\url{https://www.atmos-chem-phys.net/10/10803/2010/, 2010}.

\bibitem[{Patricola et~al.(2018)Patricola, Saravanan,, and
Chang}]{Patricola etal2018}
Patricola, C. M., Saravanan, R., and Chang, P. The response of Atlantic
tropical cyclones to suppression of African easterly waves. Geophysical
Research Letters, 45, 471- 479.

[revised manuscript text omitted]

%\bibitem[AUTHOR(YEAR)]{LABEL1}

```
%REFERENCE 1
%\bibitem[AUTHOR(YEAR)]{LABEL2}
%REFERENCE 2
\end{thebibliography}
%
% FIGURES
%
% Figure 1
%
%\newpage
%\begin{figure}[p]
%\centerline{\includegraphics[scale=0.4]{figure1}}
%\caption{A time series of the number of TC genesis events detected each
day from an idealized simulation of TC formation in a tropical channel at
a homogeneous horizontal resolution of 27 km, using the Weather Research
and Forecasting (WRF-ARW) model (Kieu et al. 2018).}\label{fig1}
%\end{figure}
%
% Figure 2
%
\newpage
\begin{figure}
\centering
\includegraphics[width=0.8\textwidth]{figure2}\\
\caption{Illustration of the zonal wind that is derived from the steady-
state flow $\psi_{S}$ in the ITCZ model \eqref{eq1} with the external
forcing given by Eq. \eqref{eq2}. The dotted curve represents the
horizontal profile of the mean flow, while the black arrows represent the
direction of the mean flow for the tropical channel domain $\Omega_a$.
The blue dashed line denotes the location of the ITCZ.} \label{fig2}
\end{figure}
%
% Figure 3
%
\newpage
\begin{figure}
\centering
\includegraphics[width=1.0\textwidth,height=0.5\textwidth]{Figure1_r1}
\caption{Marginal stability curves $R_{m}^*(a)$ obtained from the
constraint on the eigenvalue $\Re\rho_{m,1}(R)=0$ for a range of the
aspect ratio $0.1\leq a\leq0.35$.} \label{fig3}
\end{figure}
%
% Figure 4
%
\newpage
\begin{figure}
\centering
\includegraphics[width=1.0\textwidth,height=0.5\textwidth]{Figure2_r1}
\caption{The dependence of the first critical wave number $m=n$ on the
scale factor $a$, assuming the Rossby number $\epsilon = 0.5$ and the
Ekman number $E=0.05$ similar to Figure 2.}\label{fig4}
  \end{figure}
%
```

```
% Figure 5
%
\newpage
\begin{figure}
\centering
\includegraphics[width=1.1\textwidth]{Figure3 r1}\\
\caption{Illustration of the streamfunction $\psi$ for the new periodic
state on the central manifold near the critical point $R^*$ after the
dynamical transition, assuming $\epsilon=0.3$, $E=0.05$, and
$R=3.8717>R^*=3.8517$. The nondimenional period is T=2.776, which
corresponds to a physical period of $\sim $ 3.213 days. Superimposed are
corresponding vector flows derived from the streamfunction.}\label{fig5}
\end{figure}
%
% Figure 6
%
%\newpage
%\begin{figure}
%  \centering
%  \includegraphics[width=0.9\textwidth]{figure6}\\
%  \caption{Sea level pressure (shaded) distribution as obtained from a
simulation of the tropical cyclone formation using the Weather Research
and Forecasting model for the tropical channel.}\label{fig6}
%\end{figure}
%\end{document}

%% Since the Copernicus LaTeX package includes the BibTeX style file
copernicus.bst,
%% authors experienced with BibTeX only have to include the following two
lines:
%%
%% \bibliographystyle{copernicus}
%% \bibliography{example.bib}
%%
%% URLs and DOIs can be entered in your BibTeX file as:
%%
%% URL = {http://www.xyz.org/~jones/idx_g.htm}
%% DOI = {10.5194/xyz}

%% LITERATURE CITATIONS
%%
%% command                        & example result
%% \citet{jones90}|              & Jones et al. (1990)
%% \citep{jones90}|              & (Jones et al., 1990)
%% \citep{jones90,jones93}|      & (Jones et al., 1990, 1993)
%% \citep[p.~32]{jones90}|       & (Jones et al., 1990, p.~32)
%% \citep[e.g.,][]{jones90}|     & (e.g., Jones et al., 1990)
%% \citep[e.g.,][p.~32]{jones90}| & (e.g., Jones et al., 1990, p.~32)
%% \citeauthor{jones90}|         & Jones et al.
%% \citeyear{jones90}|           & 1990
```

```
%% FIGURES

%% When figures and tables are placed at the end of the MS (article in
one-column style), please add \clearpage
%% between bibliography and first table and/or figure as well as between
each table and/or figure.

%% ONE-COLUMN FIGURES

%%f
%\begin{figure}[t]
%\includegraphics[width=8.3cm]{FILE NAME}
%\caption{TEXT}
%\end{figure}
%
%%% TWO-COLUMN FIGURES
%
%%f
%\begin{figure*}[t]
%\includegraphics[width=12cm]{FILE NAME}
%\caption{TEXT}
%\end{figure*}
%
%
%%% TABLES
%%%
%%% The different columns must be seperated with a & command and should
%%% end with \\ to identify the column brake.
%
%%% ONE-COLUMN TABLE
%
%%t
%\begin{table}[t]
%\caption{TEXT}
%\begin{tabular}{column = lcr}
%\tophline
%
%\middlehline
%
%\bottomhline
%\end{tabular}
%\belowtable{} % Table Footnotes
%\end{table}
%
%%% TWO-COLUMN TABLE
%
%%t
%\begin{table*}[t]
%\caption{TEXT}
%\begin{tabular}{column = lcr}
%\tophline
%
%\middlehline
```

```
%
%\bottomhline
%\end{tabular}
%\belowtable{} % Table Footnotes
%\end{table*}
%
%%% LANDSCAPE TABLE
%
%%t
%\begin{sidewaystable*}[t]
%\caption{TEXT}
%\begin{tabular}{column = lcr}
%\tophline
%
%\middlehline
%
%\bottomhline
%\end{tabular}
%\belowtable{} % Table Footnotes
%\end{sidewaystable*}
%
%
%%% MATHEMATICAL EXPRESSIONS
%
%%% All papers typeset by Copernicus Publications follow the math
typesetting regulations
%%% given by the IUPAC Green Book (IUPAC: Quantities, Units and Symbols
in Physical Chemistry,
%%% 2nd Edn., Blackwell Science, available at:
http://old.iupac.org/publications/books/gbook/green_book_2ed.pdf, 1993).
%%%
%%% Physical quantities/variables are typeset in italic font (t for time,
T for Temperature)
%%% Indices which are not defined are typeset in italic font (x, y, z, a,
b, c)
%%% Items/objects which are defined are typeset in roman font (Car A, Car
B)
%%% Descriptions/specifications which are defined by itself are typeset
in roman font (abs, rel, ref, tot, net, ice)
%%% Abbreviations from 2 letters are typeset in roman font (RH, LAI)
%%% Vectors are identified in bold italic font using \vec{x}
%%% Matrices are identified in bold roman font
%%% Multiplication signs are typeset using the LaTeX commands \times (for
vector products, grids, and exponential notations) or \cdot
%%% The character * should not be applied as mutliplication sign
%
%
%%% EQUATIONS
%
%%% Single-row equation
%
%\begin{equation}
%
%\end{equation}
```

```latex
%
%%% Multiline equation
%
%\begin{align}
%& 3 + 5 = 8\\
%& 3 + 5 = 8\\
%& 3 + 5 = 8
%\end{align}
%
%
%%% MATRICES
%
%\begin{matrix}
%x & y & z\\
%x & y & z\\
%x & y & z\\
%\end{matrix}
%
%
%%% ALGORITHM
%
%\begin{algorithm}
%\caption{...}
%\label{a1}
%\begin{algorithmic}
%...
%\end{algorithmic}
%\end{algorithm}
%
%
%%% CHEMICAL FORMULAS AND REACTIONS
%
%%% For formulas embedded in the text, please use \chem{}
%
%%% The reaction environment creates labels including the letter R, i.e.
(R1), (R2), etc.
%
%\begin{reaction}
%%% \rightarrow should be used for normal (one-way) chemical reactions
%%% \rightleftharpoons should be used for equilibria
%%% \leftrightarrow should be used for resonance structures
%\end{reaction}
%
%
%%% PHYSICAL UNITS
%%%
%%% Please use \unit{} and apply the exponential notation

\end{document}
```

---

## Author Response (AR2)

**DEPARTMENT OF EARTH AND ATMOSPHERIC SCIENCES**

**INDIANA UNIVERSITY**
**College of Arts and Sciences**

*Date: June 14, 2019*

To Co-Editor Mathias Palm
Atmospheric Chemistry and Physics
**Ref**: Manuscript # ACP-2018-1253

Dear Dr. Mathias Palm,

We would like to thank you for your editorial assistance of our work entitled "*Large-scale Dynamics of Tropical Cyclone Formation Associated with ITCZ Breakdown*", co-authored with Quan Wang and The-Anh Vu. In this revision, we have corrected some typos that you have helped point out and modified few discussions in Section 4 to avoid confusion about the PES conditions and the dynamical transition on the central manifold. Other than these few minor modifications, there is no other change in this final version.

Please let us know if you have any further inquiry or question. We thank you again and look forward to hearing the final editorial decision from you,

Respectfully,
Chanh Kieu, Ph.D.

1001 E. Tenth Street     Bloomington, IN  47405     phone: 812-856-5704     fax (812-855-7899)